# ON INFORMATION MAXIMISATION IN MULTI-VIEW SELF-SUPERVISED LEARNING

## ABSTRACT

The strong performance of multi-view self-supervised learning (SSL) prompted the development of many different approaches (e.g. SimCLR, BYOL, and DINO). A unified understanding of how each of these methods achieves its performance has been limited by apparent differences across objectives and algorithmic details. Through the lens of information theory, we show that many of these approaches maximise an approximate lower bound on the mutual information between the representations of multiple views of the same datum. Further, we observe that this bound decomposes into a "reconstruction" term, treated identically by all SSL methods, and an "entropy" term, where existing SSL methods differ in their treatment. We prove that an exact optimisation of both terms of this lower bound encompasses and unifies current theoretical properties such as recovering the true latent variables of the underlying generative process (Zimmermann et al., 2021) or or isolating content from style in such true latent variables (Von Kügelgen et al., 2021). This theoretical analysis motivates a naive but principled objective (EntRec), that directly optimises both the reconstruction and entropy terms, thus benefiting from said theoretical properties unlike other SSL frameworks. Finally, we show EntRec achieves a downstream performance on-par with existing SSL methods on ImageNet (69.7% after 400 epochs) and on an array of transfer tasks when pre-trained on ImageNet. Furthermore, EntRec is more robust to modifying the batch size, a sensitive hyperparameter in other SSL methods.

## 1 INTRODUCTION

Representation learning commonly tackles the problem of learning compressed representations of data which capture their semantic information. A necessary, but not sufficient, property of a good representation is thus that it is highly informative of said data. For this reason, many representation learning methods aim to maximise the mutual information between the input data and the representations, while including some biases in the model that steer that information to be semantic, e.g. (Agakov, 2004; Alemi et al., 2017; Hjelm et al., 2018; Oord et al., 2018; Velickovic et al., 2019). Moreover, mutual information has been the central object to understand the performance of many of these algorithms (Saxe et al., 2019; Rodríguez Gálvez et al., 2020; Goldfeld & Polyanskiy, 2020).

A subfield of representation learning is self-supervised learning (SSL), which consists of algorithms that learn representations by means of solving an artificial task with self-generated labels. A particularly successful approach to SSL is multi-view SSL, where different views of the input data are generated and the self-generated task is to make sure that representations of one view are predictive of the representations of the other views c.f. (Jing & Tian, 2020; Liu et al., 2022).

Multi-view SSL algorithms based on the `InfoNCE` (Oord et al., 2018) like (Bachman et al., 2019; Federici et al., 2020; Tian et al., 2020a) focus on maximising the mutual information between the representations and the input data by maximising the mutual information between the representations of different views (Poole et al., 2019). Similarly, Shwartz-Ziv et al. showed that (Bardes et al., 2022, `VICReg`) also maximises this information, even though it was not designed for this purpose. Moreover, Tian et al. (2020b); Tsai et al. (2020) provide perspectives on why maximising this mutual information is attractive and discuss some of its properties. However, Tschannen et al. (2019); McAllester & Stratos (2020) warn about the caveats of this maximisation (e.g. that it is not sufficient for good representations). Here, we complement these efforts from multiple fronts and contribute:

- Showing that maximising the lower bound (1) on the mutual information between representations of different views has desirable properties in good representations (Section 2). More precisely, we show that this maximisation unifies current theories on learning the true explanatory factors of the input (Zimmermann et al., 2021) and separating semantic from irrelevant information (Von Kügelgen et al., 2021).

- Showing how many existing multi-view SSL algorithms also maximise this mutual information, although not exactly maximising the lower bound (1). This completes the picture of contrastive methods with an analysis of (Chen et al., 2020a, SimCLR), where such a result was only known for (Tian et al., 2020a, CMC)-like methods (Poole et al., 2019; Wu et al., 2020) under the InfoNCE (Oord et al., 2018) assumptions requiring i.i.d. negative samples. It also provides a unifying framework with other projections' reconstruction methods such as (Chen & He, 2021, SimSiam), (Grill et al., 2020, BYOL), (Caron et al., 2018; 2020, DeepCluster and SwAV), and (Caron et al., 2021, DINO).

- Demonstrating how a proposed naive method that directly maximises the aforementioned bound (1) on this mutual information (EntRec) has comparable performance to current state-of-the-art methods and is more robust to changes in training hyperparameters such as the batch size (Section 4 and Section 5).

This paper is a recognition of the importance of maximising the mutual information between the representations of different views of the input data, as doing so by maximising (1) has desirable properties (Section 2), and many methods that maximise it (Section 3), including naive ones (Section 4), have good empirical performance (Section 5). However, since maximising mutual information is not sufficient for good representations (Tschannen et al., 2019), this paper is also a call to include more biases in the model and the optimisation enforcing the representations to learn semantic information. Appendix A completes the positioning of the paper with respect to related work.

**Notation**   Upper-case letters $X$ represent random objects, lower-case letters $x$ their realisations, calligraphic letters $\mathcal{X}$ their outcome space, and $\mathbb{P}_X$ their distribution. Random objects $X$ are assumed to have a density $\mathsf{p}_X$ with respect to some measure $\mu$,[1] and the expectation of a function $f$ of $X$ is written as $\mathbb{E}[f(X)] := \mathbb{E}_{x \sim \mathsf{p}_X}[f(x)]$. When two random objects $X, Y$ are considered, the conditional density of $X$ given $Y$ is written as $\mathsf{p}_{X|Y}$, and for each realisation $y$ of $Y$ it describes the density $\mathsf{p}_{X|Y=y}$. Sometimes, the notation is abused to write a "variational" density $\mathsf{q}_{X|Y}$ of $X$ given $Y$. Formally, this amounts to considering a different random object $\hat{X}$ such that $\mathsf{p}_{\hat{X}|Y} = \mathsf{q}_{X|Y}$. The mutual information between two random objects $X$ and $Y$ is written as $\mathsf{I}(X; Y)$, and their conditional mutual information given the random object $Z$ as $\mathsf{I}(X; Y|Z)$. The Shannon entropy and differential entropy of a random object $X$ are both written as $\mathsf{H}(X)$, and are clear from the context. The Jensen-Shannon divergence between two distributions $\mathbb{P}$ and $\mathbb{Q}$ is written as $\mathsf{D}_{\mathsf{JS}}(\mathbb{P}\|\mathbb{Q})$. A set of $k$ elements $x^{(1)}, \ldots, x^{(k)}$ is denoted as $x^{(1:k)}$, a (possibly unordered) subsequence $x^{(a)}, \ldots, x^{(b)}$ of those elements is denoted as $x^{(a:b)}$, and all the elements in $x^{(1:k)}$ except of $x^{(i)}$ is denoted as $x^{(-i)}$.

## 2   MULTI-VIEW SSL AND MUTUAL INFORMATION

In multi-view SSL, two (or more) views (potentially generated using augmentations) of the same data sample $X$ are generated (Bachman et al., 2019; Tian et al., 2020a;b; Chen et al., 2020a; Caron et al., 2020; Zbontar et al., 2021). Views $V_1, V_2$ are engineered such that most of the semantic information $S$ of the data is preserved (Tian et al., 2020b). This process generates two branches where the views are processed to generate representations $R_1, R_2$ which are later projected into a lower dimensional space $Z_1, Z_2$. Finally, the model's parameters $\theta$ are optimised so that the projected representations (projections) from one branch, say $Z_1$, are predictive of the representations of the other branch $Z_2$ (see Figure 1). In particular, as shown in Section 3, many mutli-view SSL methods aim to maximise the mutual information between the projections $\mathsf{I}(Z_1; Z_2)$.

Consider the following decomposition of the mutual information (Agakov, 2004; Rodríguez Gálvez et al., 2020)

$$\mathsf{I}(Z_1; Z_2) = \mathsf{H}(Z_2) - \mathsf{H}(Z_2|Z_1) \geq \overbrace{\mathsf{H}(Z_2)}^{\text{Entropy}} + \overbrace{\mathbb{E}\big[\log \mathsf{q}_{Z_2|Z_1}(Z_2)\big]}^{\text{Reconstruction}}. \tag{1}$$

---

[1]Here, this measure will either be the Lebesgue measure and $\mathsf{p}_X$ will denote the standard probability density function (pdf) or the counting measure and $\mathsf{p}_X$ will denote the standard probability mass function (pmf).

The role of both terms from (1) in SSL is distinct: the entropy term determines how much information from one projection *can* be learnt, while the reconstruction term determines how much of this available information *is* learnt. For instance, imagine the projections lay on the sphere: the more spread out (higher entropy) the projections of different images are, the more revealing (higher mutual information) it is if projections from different views of the same image are close (lower reconstruction error). On the other hand, if all images of one branch are projected to the same point (lowest entropy, also known as *collapse*), the projections from the other branch can't reveal any information about them, because their location is always the same.

Although large mutual information does not necessarily imply good downstream performance (Tschannen et al., 2019), maximising this lower bound is a sensitive objective since it promotes learning the semantic information and discarding irrelevant information.

To gain intuition, assume the data can be separated into some semantic $S$ and some irrelevant variables $U$ such that $X = \varphi(S, U)$ and $S \perp\!\!\!\perp U$. Further assume that the views can be written as $V_1 = \varphi(S, U_1)$ and $V_2 = \varphi(S, U_2)$, where $U_1$ and $U_2$ independent. Since the mutual information between the views contains only semantic information, maximising $\mathsf{I}(Z_1, Z_2)$ encourages the projections to learn only semantic information. Indeed, imagine the projections contain integrally the irrelevant variables, i.e. $Z_1 = (\psi_1(S), U_1)$, $Z_2 = (\psi_2(S), U_2)$, then their mutual information would be the same as if they did not contain them at all: $\mathsf{I}(\psi_1(S), U_1; \psi_2(S), U_2) = \mathsf{I}(\psi_1(S); \psi_2(S))$. Furthermore, assume the projections lay in a compact set $\mathcal{Z} \subseteq \mathbb{R}^d$ and the reconstruction density is defined with a semi-metric $\rho$ such that $\mathsf{q}_{Z_2|Z_1=z_1}(z_2) \propto e^{-\rho(z_1, z_2)}$. Then, maximising the reconstruction term minimises $\mathbb{E}[\rho(Z_1, Z_2)]$, thus pulling together the non linear mappings $\pi_\theta \circ f_\theta \circ \varphi(S, U_1)$ and $\pi_\xi \circ f_\xi \circ \varphi(S, U_2)$ (see also Figure 1). Therefore, if the reconstruction is maximised, on average, the irrelevant variables $U_1$, $U_2$ are not contributing to $Z_1$ and $Z_2$, therefore promoting the discarding of irrelevant information.

**Remark 1.** *SSL promotes learning semantic information and discarding irrelevant information. This highlights the importance of the selection of the views of the input data. This is not the object of this paper, and it has been previously studied by Tian et al. (2020b). A similar insight can be obtained with Tsai et al. (2020)'s framework.*

Importantly, such benefits can be formalised when directly maximising the lower bound (1), as this maximisation unifies the theory from Zimmermann et al. (2021) and Von Kügelgen et al. (2021).

**Theorem 1** (Informal. Details in Appendix B)**.** *Assume there is a true data generating process $X = g(\tilde{Z})$, where $g$ is invertible and $V_1$ and $V_2$ can be understood as $g(\tilde{Z}_1)$ and $g(\tilde{Z}_2)$. Then,*

1. *Under Zimmermann et al. (2021)'s conditions, maximising (1) ensures the projections $Z$ equate the true latent variables $\tilde{Z}$ up to affine transformations.*

2. *Assume also the true latent variables are separated into semantic (or content) $S$ and irrelevant (or style) $U$ variables, such that $\tilde{Z} = [S, U]$, where $[\cdot]$ is the concatenation operator. Then, under Von Kügelgen et al. (2021)'s conditions, maximising (1) ensures that the projections isolate semantic information, in the sense that there is a bijection from $Z$ to $S$.*

## 3 MUTLI-VIEW SSL METHODS MAXIMISE MUTUAL INFORMATION

In this section, we demonstrate how many different multi-view SSL methods aim to maximise the mutual information $\mathsf{I}(Z_1, Z_2)$. In Figure 1, we schematically highlight the four prototypes that all main multi-view SSL methods that we are aware of can be partitioned into.

In the following, we exhibit how under the lens of the decomposition (1), the different methods employ different *reconstruction* densities, $\mathsf{q}_{Z_2|Z_1}$ or $\mathsf{q}_{W_2|Z_1}$, and different ways to maximise or control the entropy, $\mathsf{H}(Z_2)$ or $\mathsf{H}(W_2)$, which is empirically shown to be controlled in Appendix H. Importantly, none of them directly maximises the lower bound (1) of $\mathsf{I}(Z_1, Z_2)$, preventing them from the theoretical benefits highlighted in **Theorem 1**.

First, in Section 3.1, we study contrastive methods (Tian et al., 2020a;b; Bachman et al., 2019; Chen et al., 2020a) and later, in Section 3.2, pure latent variables' (or projections') reconstruction methods (Grill et al., 2020; Chen & He, 2021; Caron et al., 2018; 2020; 2021).

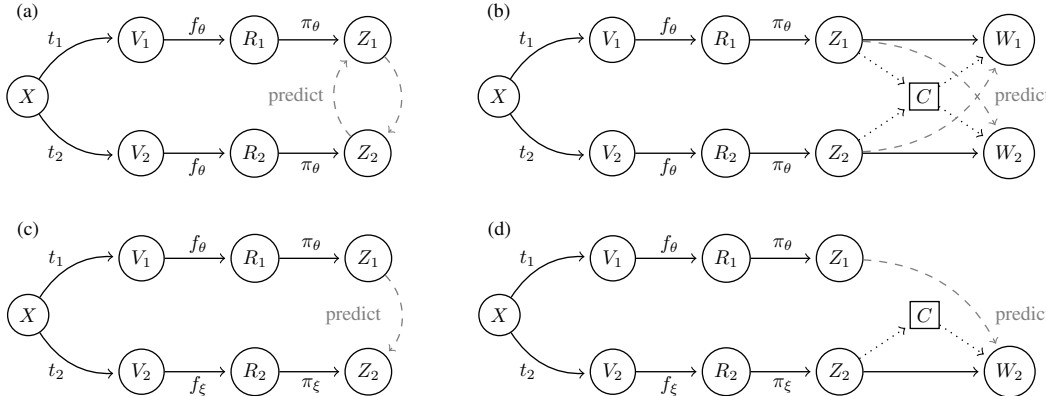

Figure 1: Graphical representation of main multi-view SSL prototypes. Solid lines describe the main variable flow: an image $X$ is transformed with augmentations $t_1$, $t_2$ to generate two views $V_1, V_2$ that are encoded into representations $R_1, R_2$ and projected into $Z_1, Z_2$ (and potentially further processed into $W_1, W_2$). Dashed lines describe method objectives, and dotted lines indicate optional relationships between variables. **Top row:** the parameters of the encoder $f$ and projector $\pi$ are shared for the processing of both views and the projections are manipulated so projections of one view are predictive of the other, and vice-versa. **Bottom row:** the parameters of the processing of $V_2$ are distinct and the projections are manipulated so that projections of $V_1$ are predictive of projections of $V_2$. **Left column:** the projections are not further processed. **Right column:** the projections are further processed into a surrogate variable $W_1, W_2$ (potentially using another variable $C$), and then are manipulated so that projections of one view are predictive of the surrogate variable of the other. For example, (a) is followed by `SimCLR` and `EntRecCont`, (b) is followed by `SwAV` and `EntRecDisc`, (c) is followed by `BYOL`, and (d) is followed by `DINO`.

## 3.1 CONTRASTIVE LEARNING METHODS

Contrastive learning methods (Wu et al., 2018; Bachman et al., 2019; Tian et al., 2020a;b; Chen et al., 2020a;b; He et al., 2020; Ramapuram et al., 2021) have the `InfoNCE` loss (Oord et al., 2018) at their core and usually have a symmetric structure (Figure 1a).

Consider a batch of $k$ data samples $X^{(1:k)}$. For each projection of each view of a sample $X^{(i)}$, say $Z_1^{(i)}$, these methods consider the projection of the other view of that image $Z_2^{(i)}$ its *positive pair* and try to identify such a pair among a set of other projections (or *negative pairs*) by minimising a cross-entropy loss based on a similarity score. This similarity score is usually defined as the temperature normalised cosine similarity $\text{sim}(\cdot, \cdot)/\tau$. Then, the different methods are essentially distinguished by the projections they consider negative pairs.

In what follows, we introduce two representatives of these methods and their relationship with maximising $\mathsf{I}(Z_1, Z_2)$. In Appendix C, we give the details and caveats of the analyses and discuss further contrastive methods, such as (He et al., 2020, `MoCo`), that inherit the analyses below.

**Contrastive Multiview Coding** In contrastive multiview coding (Tian et al., 2020a;b, `CMC`), the negative pairs of a projection from one branch, say $Z_1^{(i)}$, are all the other projections from the opposite branch $Z_2^{(-i)}$. That is, for a batch of size $k$ the optimised loss is

$$\mathcal{L}_{\text{CMC}}(\theta) := \frac{1}{k} \sum_{b=1}^{2} \sum_{i=1}^{k} \log \left( \frac{\exp\left(\text{sim}(Z_b^{(i)}, Z_{\bar{b}}^{(i)})/\tau\right)}{\sum_{j=1}^{k} \exp\left(\text{sim}(Z_b^{(i)}, Z_{\bar{b}}^{(j)})/\tau\right)} \right), \tag{2}$$

where $\bar{b}$ is the opposite branch of $b$. This loss benefits from all the properties of the `InfoNCE` (Oord et al., 2018; Poole et al., 2019). For example, if both sets of projections $Z_1^{(i)}$ as well as $Z_2^{(j)}$ are i.i.d., then minimising $\mathcal{L}_{\text{CMC}}$ maximises a lower bound on the mutual information; more precisely $\mathsf{I}(Z_1; Z_2) \geq \log k - \mathcal{L}_{\text{CMC}}(\theta)/2$. However, minimising (2) does not directly maximise the lower bound (1). Looking at (10) in Appendix C, we can see how the numerator of the logarithm in (2) is

recovered by considering the reconstruction density $q_{Z_2|Z_1=z_1}$ to be von Mises–Fisher density with mean direction $z_1$ and parameter $1/\tau$ and symmetrising with the reconstruction of the other branch. However, the denominator of the logarithm is not a Joe (1989)'s kernel density estimator (KDE) approximation of the entropy $H(Z_2)$ since the average of the logarithm of the kernel is taken over samples of $\mathbb{P}_{Z_1}$ and not $\mathbb{P}_{Z_2}$. Hence, it only maximises an estimation of the entropy if $Z_1$ and $Z_2$ have the same (or approximately the same) marginals, i.e. $\mathbb{P}_{Z_1} \cong \mathbb{P}_{Z_2}$.

**SimCLR**   In SimCLR (Chen et al., 2020a), the negative pairs of a projection from one branch, say $Z_1^{(i)}$, are all the other projections $Z_1^{(-i)}, Z_2^{(1:k)}$. That is, for a batch of size $k$ the optimised loss is

$$\mathcal{L}_{\text{SimCLR}}(\theta) := \frac{1}{2k} \sum_{b=1}^{2} \sum_{i=1}^{k} \log \left( \frac{\exp\left(\text{sim}(Z_b^{(i)}, Z_{\bar{b}}^{(i)})/\tau\right)}{\sum_{b'=1}^{2} \sum_{j=1}^{k} \mathbb{I}\left((i,b) \neq (j,b')\right) \exp\left(\text{sim}(Z_b^{(i)}, Z_{b'}^{(j)})/\tau\right)} \right). \quad (3)$$

This loss does not directly inherit the InfoNCE properties as the CMC loss (see Appendix **C.2.1**). However, (3) can be approximately rewritten according to the decomposition (1), as is demonstrated in (13) in Appendix **C.2.2**. The numerator of the logarithm in (3) is then recovered using a von Mises–Fisher reconstruction density, but the denominator is not an approximation of the entropy since the average of the logarithm of the kernel is taken, this time, over samples of both $\mathbb{P}_{Z_1}$ and $\mathbb{P}_{Z_2}$. Nonetheless, this difference allows us to consider that the samples come from the mixture $\mathbb{P}_Z = 0.5\mathbb{P}_{Z_1} + 0.5\mathbb{P}_{Z_2}$, and thereby recover the KDE estimator from Joe (1989) of $H(Z)$ with a von Mises–Fisher density kernel.[2] Therefore, taking into account the relationship between the Jensen–Shannon's divergence and the entropies of two random variables results in the approximate inequality

$$I(Z_1; Z_2) \gtrsim \log k - \mathcal{L}_{\text{SimCLR}}(\theta) - D_{\text{JS}}(\mathbb{P}_{Z_1} \| \mathbb{P}_{Z_2}),$$

which reveals that minimising (3) approximately maximises $I(Z_1; Z_2)$ when $\mathbb{P}_{Z_1}$ and $\mathbb{P}_{Z_2}$ are equal.

## 3.2   PROJECTIONS' RECONSTRUCTION METHODS

The projections' reconstruction methods (Grill et al., 2020; Chen & He, 2021; Caron et al., 2020; 2021) focus on making sure that predictions from one branch are informative of those from the other branch. To achieve this goal, their loss functions consist of a term that can be understood as the reconstruction term in (1) with the appropriate density.

To avoid collapse in the absence of negative pairs, they have to employ different engineering techniques that as we show can help to maintain a high entropy term in (1) in different ways. Below we analyse the self-distillation methods (Grill et al., 2020, BYOL) and (Caron et al., 2021, DINO) from our information-theoretic viewpoint based on (1). In Appendix **D**, we further analyse other (non self-distillation) projections' reconstruction methods such as (Chen & He, 2021, SimSiam) and (Caron et al., 2018; 2020, DeepCluster and SwAV). While DeepCluster always keeps a certain level of entropy and SwAV maximises it, the other methods rely on parameters to maintain it.

**BYOL**   In Bootstrap Your Own Latent (Grill et al., 2020, BYOL), they consider an asymmetric structure (Figure 1c) and try to predict the projections from the bottom branch $Z_2$ using the predictions of the top branch $Z_1$ and a small predictor network $g_\theta$. For this purpose, they try to minimise the $\ell_2$ normalised mean squared error,

$$\mathcal{L}_{\text{BYOL}}(\theta) := \frac{1}{k} \sum_{i=1}^{k} \left\| \overline{g_\theta(Z_1^{(i)})} - \overline{Z_2^{(i)}} \right\|^2 = 2 \left( 1 - \frac{1}{k} \sum_{i=1}^{k} \text{sim}\left(g_\theta(Z_1^{(i)}), Z_2^{(i)}\right) \right)$$

using gradient descent, where $\bar{a} := a/\|a\|$. Note that this is equivalent, up to constants that do not affect the optimisation, to maximising the reconstruction term in the decomposition (1) with a von Mises–Fisher reconstruction density with mean direction $\overline{g_\theta(Z_1)}$ and concentration parameter 1, i.e., $q_{Z_2|Z_1=z_1}(z_2) \propto \exp\left(\text{sim}(g_\theta(z_1), z_2)\right)$. Note that the parameters of the branch that outputs the predicted projections $Z_2$ are parameterised with different parameters $\xi$. Hence, if these parameters were fixed or modified so that $H(Z_2)$ is increasing or maintained constant, then minimising $\mathcal{L}_{\text{BYOL}}$

---

[2]In fact, the density is Gaussian, but it essentially turns into an unnormalised von Mises–Fisher after noting all samples employed in the estimation have norm 1.

would indeed maximise the mutual information $I(Z_1; Z_2)$. Finding a way to fix or modify them in such a way is however challenging. For example, fixing $\xi$ to random values ensures constant entropy $H(Z_2)$, but also means that the then random projection $Z_2$ contains very little information about $X$. Thus, in this case, while minimising $\mathcal{L}_{\text{BYOL}}$ would maximise $I(Z_1; Z_2)$, the information learned is still little as by the data processing inequality $I(Z_1; Z_2) \leq I(X; Z_2)$. On the other hand, if $\xi$ depends on $\theta$, there is the risk of collapse: for example, in the extreme case of $\xi = \theta$, minimising $\mathcal{L}_{\text{BYOL}}$ will maximise $-H(Z_2|Z_1)$, and an optimal solution $\theta^\star$ could be a highly concentrated or degenerate $Z_1$ and $Z_2$ around one point $z$, under which $H(Z_2) \to -\infty$, which clearly does not maximise $I(Z_1; Z_2)$.

In BYOL, they circumvent these issues by updating the parameters $\xi$ during the optimisation with the moving average $\xi \leftarrow \lambda\xi + (1 - \lambda)\theta$ for some $\lambda \in (0, 1)$ close to 1. The idea (hypothesis) is two-fold: on the one hand, while $\xi$ does depend on $\theta$, the dependence is weak enough so that $H(Z_2)$ is not degrading to values yielding trivial bounds; and on the other hand, the dependence of $\xi$ on $\theta$, while weak, still makes sure that the representations $Z_2$ capture information about the data. This hypothesis is backed up by the results sweeping the parameter $\lambda$ in (Grill et al., 2020). In fact, it has later been seen (Caron et al., 2021) that this dependence resembles a Polyak-Ruppert averaging with exponential decay (Polyak & Juditsky, 1992; Ruppert, 1988), which is standard practice to improve the performance of the model, e.g. (Jean et al., 2014).

**DINO** Caron et al. (2021, DINO) also consider an asymmetric structure (Figure 1d) and, similarly to DeepCluster and SwAV, generate a discrete surrogate variable $W_2 = \phi(Z_2)$ and try to minimise a cross entropy term. More precisely, they minimise

$$\mathcal{L}_{\text{DINO}}(\theta) := \frac{1}{k}\sum_{i=1}^{k} \mathsf{s}\big((Z_2^{(i)} - C)/\tau_2\big)^\mathsf{T} \log \mathsf{s}\big(Z_1^{(i)}/\tau_1\big),$$

where $C$ is some centring variable, $\tau_1, \tau_2$ are temperature hyperparameters, and $\mathsf{s}$ is the softmax operator. Letting $\mathsf{p}_{W_2|Z_2=z_2} = \mathsf{s}\big((z_2 - C)/\tau_2\big)$ and $\mathsf{q}_{W_2|Z_1=z_1} = \mathsf{s}(z_1/\tau_1)$ shows that minimising $\mathcal{L}_{\text{DINO}}$ directly maximises the reconstruction term in the decomposition (1) of $I(Z_1, W_2) \leq I(Z_1, Z_2)$.

DINO does not directly maximise the entropy $H(W_2)$ to avoid collapse. However, they promote a high conditional entropy $H(W_2|Z_2)$ through the centring before the softmax operation defining $\mathsf{p}_{W_2|Z_2}$. To be precise, the centre $C$ is updated with a moving average of the previous projections, that is, $C \leftarrow \mu C + \frac{(1-\mu)}{k}\sum_{i=1}^{k} Z_2$ for some $\mu \in (0, 1)$. Then, the right balance between the moving average and temperature parameters $\mu$ and $\tau_2$ adjusts how uniform the conditional density $\mathsf{p}_{W_2|Z_2}$ should be. Hence, since $H(W_2|Z_2) \leq H(W_2)$, controlling the conditional entropy controls $H(W_2)$.

Finally, similarly to BYOL, DINO faces the potential problem of obtaining useless representations due to uninformative targets if the parameters $\xi$ do not ensure that the projections $Z_2$ capture enough information about the data $X$. They solve this issue as in BYOL updating them with a moving average $\xi \leftarrow \lambda\xi + (1 - \lambda)\theta$ for some $\lambda \in (0, 1)$. As previously mentioned, with the appropriate selection of $\lambda$, this resembles a Polyak-Ruppert averaging with exponential decay (Polyak & Juditsky, 1992; Ruppert, 1988) and makes sure that $Z_2$ captures information about the data $X$ (Caron et al., 2021).

## 4 THE ENTREC METHOD

Previously, we established how many multi-view SSL methods aim to maximise the mutual information between the projections on both branches $I(Z_1; Z_2)$, and that they can be understood by decomposing the mutual information into an entropy and a reconstruction term as in (1). However, none of these methods takes such a decomposition and tries to maximise these two terms directly.

In this section, we present the EntRec method which does exactly that, and naively maximises both entropy and reconstruction terms. The method comes in two variants: (i) EntRecCont, a direct maximisation of $I(Z_1, Z_2)$, where the entropy is estimated with a KDE, that follows Figure 1a's structure; or (ii) EntRecDisc, a generation of a discrete surrogate random variable $W_b = \phi(Z_b)$ and posterior maximisation of $\big(I(Z_1; W_2) + I(W_1; Z_2)\big)/2 \leq I(Z_1; Z_2)$, where the entropy of $W_b$ can be easily estimated with a plug-in estimator, that follows Figure 1c's structure. EntRecCont, unlike all of the methods described previously, directly maximises the lower bound (1) on the mutual information $I(Z_1; Z_2)$, and therefore enjoys the theoretical properties from **Theorem 1**, such as recovering true latent variables and separating semantic from irrelevant information. However, the KDE requires a large number of samples to properly estimate the entropy. EntRecDisc addresses this potential drawback by estimating the entropy of a discrete (surrogate) random variable

instead, although at the price of maximising a lower bound on $I(W_1; Z_2)$ only. Hence, it maximises a looser bound on $I(Z_1, Z_2)$ and does not benefit from the theoretical properties of **Theorem 1**. The pseudocode of the algorithm is given in Appendix **E**.

### 4.1 ENTRECCONT

`EntRecCont` considers the mutual information decomposition from (1) and directly maximises an estimate of the lower bound by maximising the loss function

$$\mathcal{L}_{\text{EntRecCont}}(\theta) \coloneqq -\frac{1}{2} \sum_{b=1}^{2} \left( \hat{\mathsf{H}}(Z_b, Z_b^{(1:k)}) + \frac{1}{k} \sum_{i=1}^{k} \log \left( \mathsf{q}_{Z_b|Z_{\bar{b}}=Z_{\bar{b}}^{(i)}}(Z_b^{(i)}) \right) \right), \qquad (4)$$

where $\hat{\mathsf{H}}(Z_b; Z_b^{(1:k)})$ is an estimate of the entropy and $\mathsf{q}_{Z_b|Z_{\bar{b}}}$ is a parameterised reconstruction density. Multiple potential estimates of the entropy $\mathsf{H}(Z_b)$ exist, but this paper considers those generated with Joe (1989)'s KDE, which comes with the same caveats mentioned for the analysis of contrastive methods in Appendix **C.3**. More precisely, the estimator takes the form

$$\hat{\mathsf{H}}(Z_b, Z_b^{(1:k)}) = -\frac{1}{k} \sum_{i=1}^{k} \log \left( \frac{1}{kh^d} \sum_{j=1}^{k} \mathsf{q}\left( \frac{Z_b^{(i)} - Z_b^{(j)}}{h} \right) \right),$$

where $\mathsf{q}$ is some kernel density and $h \in \mathbb{R}_+$ is its bandwidth.

The loss (4) recovers Wang & Isola (2020)'s alignment-uniformity loss up to constants independent of the parameters $\theta$ when the reconstruction density is $\mathsf{q}_{Z_b|Z_{\bar{b}}=z_{\bar{b}}}(z_b) \propto \exp\left( -\|z_b - z_{\bar{b}}\|^\alpha \right)$ and the kernel density is Gaussian. Applying the log-sum inequality to the entropy estimation term fully recovers that loss, revealing it is an estimation of a looser bound of the mutual information $I(Z_1, Z_2)$.

Moreover, `EntRecCont` enjoys the following theoretical benefits on its asymptotic behaviour.

**Theorem 2** (The `EntRec` loss (4) tends to a proper bound on $I(Z_1; Z_2)$)**.** *If $X_i$ are i.i.d. for all $i \in [k]$ and $f$ and $\pi$ do not use batch normalisation, then, for a constant bandwidth $h > 0$*

$$\lim_{k \to \infty} \mathcal{L}_{\text{EntRecCont}}(\theta) - C_{\text{KDE}}(h, k) = -\frac{1}{2} \sum_{b=1}^{2} \left( \mathsf{H}(Z_b) + \mathbb{E}\left[ \log \left( \mathsf{q}_{Z_b|Z_{\bar{b}}}(Z_b) \right) \right] \right) \geq -I(Z_1; Z_2), \; (5)$$

*where $C_{\text{KDE}} \in \mathcal{O}(h^4)$ and the convergence rate is detailed in Appendix **F.1**. Moreover, if $h \in \mathcal{O}(k^{-\frac{1}{d+\varepsilon}})$ for some small $\varepsilon > 0$ and $d > 4$, then (5) still holds and $C_{\text{KDE}}(h, k) \to 0$.*

This theorem reveals that as the batch size increases, `EntRecCont` maximises exactly the lower bound (1) and therefore it enjoys the theoretical properties highlighted in Section **2** (details in Appendix **B**): (i) if there exists a true generative process that generates the data $X$ from some true latent variables, maximising (4) may recover these latent variables; and (ii) if these latent variables are separated into some semantic (or content) variables and some irrelevant (or style) variables, maximising (4) may isolate and recover the content latent variables.

### 4.2 ENTRECDISC

The KDE of the entropy converges slowly for high dimensions (c.f. **Theorem 2**), which requires large batch sizes. It is also computationally costly (requires $\mathcal{O}(k^2d)$ operations like other constrastive methods such as `CMC` or `SimCLR`, as seen comparing (10) and (13) with (4)) for large batch sizes. Therefore, it is suitable to have an alternative to (4) that does not involve KDEs, for example considering discrete random variables instead of continuous ones.

`EntRecDisc` considers a discrete surrogate random variable $W_b = \phi(Z_b)$ and maximises a lower bound of $\left( I(Z_1; W_2) + I(Z_2; W_1) \right)/2$. This way, (i) considering the lower bound from (1) to each of the terms allows us to deal with the entropy of a discrete random variable and avoid KDEs, and (ii) by the data processing inequality, we are still maximising a lower bound on $I(Z_1, Z_2)$. To be precise, this version of `EntRec` minimises the loss function

$$\mathcal{L}_{\text{EntRecDisc}}(\theta) \coloneqq -\frac{1}{2} \sum_{b=1}^{2} \left( \hat{\mathsf{H}}(W_b; W_b^{(1:k)}) + \frac{1}{k} \sum_{i=1}^{k} \mathbb{E}\left[ \log \left( \mathsf{q}_{W_b|Z_{\bar{b}}=Z_{\bar{b}}^{(i)}}(W_b^{(i)}) \right) \right] \right). \qquad (6)$$

At first sight, both (4) and (6) seem indistinguishable except from the fact that now $W_b$ is replacing $Z_b$ and that the loss is a looser bound on $\mathsf{I}(Z_1; Z_2)$. However, this extra processing of the projections allows us to estimate the entropy better and to calculate the reconstruction term (cross-entropy) exactly. For instance, let $\mathsf{p}_{W_b|Z_b=z_b} = \mathsf{s}(z_b)$, where $\mathsf{s}$ is the softmax operator. Then, $W_b$ is a discrete random variable in $[d]$ and $\mathsf{H}(W_b)$ may be estimated with the plug-in estimator using the empirical estimate of the marginal, i.e. $\hat{\mathsf{p}}_{W_b} = \frac{1}{k} \sum_{i=1}^{k} \mathsf{p}_{W_b|Z_b=Z_b^{(i)}}$. Finally, letting the reconstruction density be $\mathsf{q}_{W_b|Z_{\bar{b}}=z_{\bar{b}}} = \mathsf{s}(z_{\bar{b}})$ results in the following loss function

$$\mathcal{L}_{\text{EntRecDisc}}(\theta) = -\frac{1}{2k} \sum_{b=1}^{2} \sum_{i=1}^{k} \left( -\mathsf{s}(Z_b^{(i)})^{\intercal} \log\left(\frac{1}{k}\sum_{j=1}^{k} \mathsf{s}(Z_b^{(j)})\right) + \mathsf{s}(Z_b^{(i)})^{\intercal} \log\left(\mathsf{s}(Z_{\bar{b}}^{(i)})\right) \right). \quad (7)$$

As opposed to the KDE of the entropy, the plug-in estimate only requires $\mathcal{O}(kd)$ operations and converges faster than the KDE (Appendix F.2). Thus equation (7) has a better asymptotic behaviour than (4). This is formalised in the following theorem.

**Theorem 3** (The `EntRec` loss (7) tends to a proper bound on $\mathsf{I}(Z_1; Z_2)$). *If $X_i$ are i.i.d. for all $i \in [k]$ and $f$ and $\pi$ do not use batch normalisation, then*

$$\lim_{k\to\infty} \mathcal{L}_{\text{EntRecDisc}}(\theta) = -\frac{1}{2} \sum_{b=1}^{2} \left( \mathsf{H}(W_b) + \mathbb{E}\left[ \log\left(\mathsf{q}_{W_b|Z_{\bar{b}}}(W_b)\right) \right] \right) \geq -\mathsf{I}(Z_1; Z_2), \quad (8)$$

*where the convergence rate is detailed in Appendix F.2.*

## 5 EXPERIMENTS

While `EntRec` has a principled derivation that allows for direct maximisation of the lower bound in (1) and as a result is equipped with desirable theoretic properties, its practical use has yet to be investigated. We do this in a series of experiments, in all of which we use ImageNet (Deng et al., 2009) for pre-training. We compare both variants of `EntRec` with all methods analysed in Section 3. All experimental details can be found in Appendix G.

In the main part of the paper, we analyse how `EntRec` compares to the other methods in terms of top-1 classification accuracy on the ImageNet test set under the linear evaluation protocol, and further study how changes in individual training hyperparameters affect this metric for each method if no other training or model hyperparameter is changed. In Appendix H, we further include a comparison in terms of transfer learning performance (fine-tuned top-1 classification accuracy on 10 natural image data sets that differ from ImageNet) and a qualitative analysis of the behaviour of the entropy term during training for `EntRecDisc` and `DINO` (for which the entropy can be estimated accurately as they use discrete surrogate variables).

As can be seen from the first column of Table 1 and Table 2, `EntRec`'s performance on the ImageNet test set is comparable to that of the contrastive methods. The same can be observed with respect to its transfer learning performance in the additional experiments that are included in Appendix H. [3]

The projections' reconstruction methods (`DINO`, `BYOL`) perform better on their optimised settings, but can struggle under changes in hyperparameters (without further adjustments). Here, we see that `EntRec` compares overall favourably in terms of robustness to changes in training hyperparemeters:

- *Batch size.* It is known that lowering the batch size can adversely affect the performance of SSL methods (Chen et al., 2020a). In Table 1 we see this is also true for `EntRec`, however to a lesser extent than for all other methods. Importantly, the projections' reconstruction methods (`DINO`, `BYOL`) which rely on engineering techniques to maintain high entropy can be very severely affected by lower batch sizes, which potentially call for further hyperparameter adjustments to recover performance. [4]

---

[3] All results are obtained with our implementation of the algorithms. The hyperparameters have been optimised by us to obtain the best performance of each of the SSL methods. For `EntRec`, we just used the parameters from `SimCLR` without any further optimisation.

[4] For example, Grill et al. (2020) use gradient accumulation so `BYOL` can cope with lower batch sizes.

- *Epochs.* Furthermore, SSL methods typically need a very high number of total training epochs to achieve their strongest performances (Grill et al., 2020). In Table 2 we see this is also true for `EntRec`. However, this time another method, `DINO`, seems to be the least affected by a lower number of total training epochs. Still, `EntRec` performs comparably to the remaining SSL methods and is notably more stable than `BYOL` again.

Altogether, these experimental results showcase that `EntRec` can indeed have practical use as a good objective for multi-view SSL beyond its theoretical benefits highlighted in the previous sections.

Table 1: Performance of the different methods across batch sizes after 400 epochs of training.

|  | Accuracy | ΔAccuracy wrt. 4096 | | |
|---|---|---|---|---|
|  | 4096 | 2048 | 1024 | 512 |
| SimCLR | 69.5 | -3.1 | -2.5 | -4.5 |
| CMC | 69.1 | -1.1 | -2.9 | -4.8 |
| BYOL | **73.8** | -2.1 | -22.0 | -42.5 |
| DINO | 71.6 | **+0.5** | -15.0 | -67.0 |
| EntRecCont | 69.7 | -0.7 | -2.3 | -4.2 |
| EntRecDisc | 66.9 | -1.0 | **-2.2** | **-3.9** |

Table 2: Performance of the different methods across epochs with batch size of 4096.

|  | Accuracy | ΔAccuracy wrt. 400 | | |
|---|---|---|---|---|
|  | 400 | 300 | 200 | 100 |
| SimCLR | 69.5 | -1.0 | -2.2 | -5.4 |
| CMC | 69.1 | -0.6 | -1.8 | -5.6 |
| BYOL | **73.8** | -1.8 | -5.0 | -13.4 |
| DINO | 71.6 | **+0.2** | **+0.3** | **-1.2** |
| EntRecCont | 69.7 | -0.8 | -2.1 | -5.9 |
| EntRecDisc | 66.9 | -0.2 | -1.1 | -4.1 |

## 6 CONCLUSION

We provided a unifying information theoretic analysis of common SSL methods, showing how they partially or approximately maximise a lower bound to the mutual information between the learned representations of distinct views of the same datum. Based on this analysis, we introduced `EntRec`, a simple SSL method that directly maximises this lower bound. We showed that it possesses a range of desirable theoretical properties, such as recovering the true latent variables of the underlying generative process or isolating content from style in such true latent variables. Furthermore, we demonstrated empirically that its classification performance is comparable to the existing SSL methods and most notably robust to changes in individual training hyperparameters such as the batch size or the number of training epochs.

**Limitations and future directions** Maximising mutual information is not enough to learn good representations, and strong inductive biases are important. For instance, the usage of certain reconstruction densities or projection spaces when maximising the lower bound (1) grants the theoretical properties highlighted in **Theorem 1** and deepened in **Appendix B**. Now that we know these methods maximise the mutual information between the representations of different views, the next step is to (i) understand which inductive biases they possess to justify their difference in performance (e.g. why `DINO` performs better when properly tuned), and (ii) design methods from first principles that both maximise this mutual information and have these inductive biases.

**Reproducibility statement** Regarding our theoretical results, we made an effort to give clear explanations of any assumptions and complete proofs of all claims in the main part of the paper in Appendices B, C, D, and F. Regarding our experimental results, we included a section in the Appendix G that clearly states the experimental protocol used to obtain these results. Furthermore, we are working to release the code used in this paper as soon as possible.

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

# Appendix

## A    RELATED WORK

This section of the Appendix puts into perspective this paper with respect to the relevant work closest to ours. First, it does so with other works that study (multi-view) SSL around mutual information, namely (Tian et al., 2020a; Tsai et al., 2020; Federici et al., 2020). Then, it continues with other works (Poole et al., 2019; Shwartz-Ziv et al.) that analysed the mutual information that particular SSL algorithms, InfoNCE-based and (Bardes et al., 2022, VICReg), is maximising. It finalises describing how the relationship between Zimmermann et al. (2021) and Von Kügelgen et al. (2021) is expanded in this work.

### A.1    MUTUAL INFORMATION-BASED FRAMEWORKS FOR MULTI-VIEW SSL

Tian et al. (2020b) focus their study on how to generate the views in multi-view SSL so that the learned representations have good downstream performance. They focus their analysis assuming a contrastive learning algorithm that maximises the InfoNCE (Oord et al., 2018) lower bound on the mutual information between the views (and more precisely on the projections of said views). They conclude that the views should share the information of a desired downstream task and nothing else, and provide with an algorithm to enforce that. Then, Tsai et al. (2020) consider multi-view SSL algorithms where one view is always the unperturbed input and show that, under certain conditions, these methods can keep task-relevant and discard task-irrelevant information for some downstream task. This is similar to (Federici et al., 2020), where they show a similar conclusion for a multi-view version of the information bottleneck (Tishby et al., 2000).

Instead, in Section 2 of this paper, we assume a perfect view selection to gain intuition on how maximising the mutual information between the representations of the different views of the input data encourages the learning of semantic information and the discarding of irrelevant information. After that, we show that under the conditions of Zimmermann et al. (2021) and Von Kügelgen et al. (2021), their theory is unified when maximising the lower bound (1) on this mutual information. More precisely, saying that the maximisation of the mutual information between the representations of different views of the same input datum can recover the true explanatory factors of the input data and separate semantic from irrelevant information.

### A.2    STUDY OF SPECIFIC ALGORITHMS

Poole et al. (2019) showed that InfoNCE maximised a lower-bound on the mutual information, which includes InfoNCE-based methods such as the CMC. This analysis was complemented by Wu et al. (2020) that showed that as long as the negative samples are i.i.d., they may come from a diffrent distribution than the positive samples. Shwartz-Ziv et al. show that (Bardes et al., 2022, VICReg) maximises the mutual information between the representations of one view and the other views. Similarly, we show how many multi-view SSL methods maximise the mutual information between the representations of different views of the input data. More precisely, we lift the i.i.d. assumption for contrastive methods and include an analysis of (Chen et al., 2020a, SimCLR), completing the picture of contrastive learning methods, as well as analysis for other and latent variables' reconstruction methods like (Chen & He, 2021, SimSiam), (Grill et al., 2020, BYOL), (Caron et al., 2018; 2020, DeepCluster and SwAV), and (Caron et al., 2021, DINO).

### A.3    RELATIONSHIP BETWEEN RECOVERING TRUE LATENT VARIABLES AND SEPARATING CONTENT FROM STYLE

Von Kügelgen et al. (2021) already discussed some connections of their work and (Zimmermann et al., 2021). In particular, they mentioned that their equation (1) could be interpreted as an alignment (numerator) and uniformity (denominator) terms, and that the latter constituted a nonparametric estimator of the entropy as the batch size grew to infinity. Therefore, they related their work with (Wang & Isola, 2020) and by proxy to (Zimmermann et al., 2021).

However, this relationship is not direct. The practical implementation of the uniformity term in (Zimmermann et al., 2021) is equivalent to a KDE estimation of the entropy, and tends to the

entropy at the rate stated in our **Theorem 2**. In Von Kügelgen et al. (2021), they consider the real entropy for their theorem, and not an approximation. So the connection in (Von Kügelgen et al., 2021) with (Zimmermann et al., 2021) only holds in the asymptotic regime.

Conversely, we prove that both results hold when maximising the lower bound (1) on the mutual information, without using the uniformity loss but the entropy directly. That is, both results are unified through mutual information maximisation. Therefore, the connection in our analysis holds without the need of employing neither the uniformity loss nor a KDE estimation of the entropy.

## B MAXIMISING MUTUAL INFORMATION PROPERTIES

This section of the Appendix formalises and contextualises **Theorem 1**'s statement.

### B.1 RECOVERING THE TRUE LATENT VARIABLES

Let us consider the standard assumption in independent components analysis (ICA), namely that the data $X$ is generated by a nonlinear, invertible generative process $X = g(\tilde{Z})$ from some original latent variables $\tilde{Z}$. Assume further that the different views from the image can be understood as $V_1 = g(\tilde{Z}_1)$ and $V_2 = g(\tilde{Z}_2)$, where there is some joint density of the latent variables $\mathsf{p}_{\tilde{Z}_1, \tilde{Z}_2}$. The next theorem shows how Zimmermann et al. (2021) theory can be adapted to prove that mutli-view SSL methods that maximise the mutual information between their projections $\mathsf{I}(Z_1; Z_2)$ can obtain projections equivalent to the true latent variables up to affine transformations.

**Theorem 4.** *Assume that the latent variables and the network's projections lay on a convex body $\mathcal{Z} \in \mathbb{R}^d$. Further assume that the latent variables' marginal distribution is uniform and that the conditional density is described by a semi-metric $\rho$ as $\mathsf{p}_{\tilde{Z}_2 | \tilde{Z}_1 = \tilde{z}_1}(\tilde{z}_2) = C(\tilde{z}_1) e^{-\rho(\tilde{z}_1, \tilde{z}_2)}$. Now let the reconstruction density match the conditional density up to a constant scaling factor $\mathsf{q}_{Z_2 | Z_1 = z_1}(z_2) = C_h(\tilde{z}_1) e^{-\alpha \rho(z_1, z_2)}$. If the generative process $g$ and the parameterised network functions $\pi \circ f$ are invertible and differentiable, and the parameters $\theta$ maximise the lower bound (1) of the mutual information $\mathsf{I}(Z_1; Z_2)$, then the projections are equivalent to the true latent variables up to affine transformations.*

*Proof.* As in (Zimmermann et al., 2021), let $h = \pi \circ f \circ g$ be a helper function that brings the true latent variables to the projections so that $Z_1 = h(\tilde{Z}_1)$ and $Z_2 = h(\tilde{Z}_2)$.

Disregard for a moment the entropy term $\mathsf{H}(Z_2)$. From (Zimmermann et al., 2021, Proposition 4) we know that if the reconstruction term is maximised (the cross entropy is minimised) then $\rho(\tilde{z}_1, \tilde{z}_2) = \alpha \rho(h(\tilde{z}_1), h(\tilde{z}_2))$ and $C(\tilde{z}_1) = C_h(\tilde{z}_1)$. Moreover, from (Zimmermann et al., 2021, Theorem 4) we further know that $h$ is an invertible affine transformation; i.e. $Z_2 = A\tilde{Z}_2 + b$ for some $A \in \mathbb{R}^{d \times d}$ and some $b \in \mathbb{R}^d$.

Now note that

$$\mathsf{H}(Z_2) = -\mathbb{E}\Big[ \log \mathbb{E}\Big[ C_h(\tilde{Z}_1) e^{-\alpha \rho(h(\tilde{Z}_1), h(\tilde{Z}_2))} \Big] \Big]$$
$$= -\mathbb{E}\Big[ \log \mathbb{E}\Big[ C(\tilde{Z}_1) e^{-\rho(\tilde{Z}_1, \tilde{Z}_2)} \Big] \Big] = \mathsf{H}(\tilde{Z}_2).$$

Then, since the latent variables' are uniformly distributed, their entropy is maximal $\mathsf{H}(\tilde{Z}) = \log |\mathcal{Z}|$.

Therefore, the unique family of maximisers of the reconstruction term recover the latent variables up to affine transformations are maximisers of the entropy, and hence are the unique family of maximisers of the mutual information. Indeed, take some other maximiser of the entropy, if it is not from this family, it is not a maximiser of the reconstruction and therefore the resulting mutual information is lower. □

**Remark 2.** *Following the same reasoning and supporting on Zimmermann et al. (2021)'s theory, we may note that in the particular case that the semi-metric $\rho$ is described by an $L^p$ norm, then the projections are equivalent to the true latent variables up to generalised permutations; that is, $Z = A\tilde{Z}$ for some $A \in \mathbb{R}^{d \times d}$ such that $(Az)_i = \alpha \beta_i z_{\sigma(i)}$, where $\alpha \in \mathbb{R}$, $\beta_i \in \{1, -1\}$, and $\sigma$ is a*

*permutation. Similarly, in the more restrictive case that the projections are in the sphere $\mathcal{Z} = \mathbb{S}^{d-1}$ and the conditional densities are von Mises–Fisher densities, then the projections are equivalent to the true latent variables up to linear transformations; that is, $Z = A\tilde{Z}$ for some $A \in \mathbb{R}^{d \times d}$ such that $A^\intercal A = \alpha I$ for some $\alpha \in \mathbb{R}$.*

### B.2 ISOLATING SEMANTIC FROM IRRELEVANT INFORMATION

Similarly to Appendix B.1, let us consider that the data $X$ is generated by a nonlinear, invertible generative process $X = g(\tilde{Z})$ from some original latent variables $\mathcal{Z}$ and that the different views can be understood as $V_1 = g(\tilde{Z}_1)$ and $V_2 = g(\tilde{Z}_2)$, where there is some joint density of the latent variables $\mathsf{p}_{\tilde{Z}_1, \tilde{Z}_2}$.

Assume that the latent variables can be written as $\tilde{Z} = [S, U]$, where $S \in \mathbb{R}^d$ is some semantic (or content) variable, $U \in \mathbb{R}^{d_u}$ is some irrelevant (or style) variable, and $[\cdot]$ denotes the concatenation operation. Furthermore, let us adopt the assumptions from Von Kügelgen et al. (2021) for the content-preserving conditional density $\mathsf{p}_{\tilde{Z}_2 | \tilde{Z}_1}$.

**Assumption 1** (Content-invariance). *The conditional density $\mathsf{p}_{\tilde{Z}_2 | \tilde{Z}_1}$ of the latent variables of different views has the form*

$$\mathsf{p}_{\tilde{Z}_2 | \tilde{Z}_1 = \tilde{z}_1}(\tilde{z}_2) = \delta(s_2 - s_1)\mathsf{p}_{U_2 | U_1 = u_1}(u_2),$$

*for all $\tilde{z}_1 = [s_1, u_1]$ and $\tilde{z}_2 = [s_2, u_2]$ in $\mathcal{Z}$ and where $\mathsf{p}_{U_2 | U_1 = u_1}$ is continuous for all $u_1 \in \mathbb{R}^{d_u}$.*

**Assumption 2** (Style changes). *Let $\mathcal{A}$ be the set of subsets of irrelevant variables $A \subseteq \{1, \ldots, d_u\}$ and let $\mathsf{p}_A$ be a density on $\mathcal{A}$. Then, the conditional density $\mathsf{p}_{U_2 | U_1}$ is obtained via*

$$a \sim \mathsf{p}_A, \quad \mathsf{p}_{U_2 | U_1, A = u_1, a}(u_2) = \delta(u_{2, a^c}, u_{1, a^c})\mathsf{p}_{U_{2, a} | U_{1, a} = u_{1, a}}(u_{2, a})$$

*where $\mathsf{p}_{U_{2, a} | U_{1, a} = u_{1, a}}$ is a continuous density for all $u_{1, a} \in \mathbb{R}^{|a|}$, and where $u_{2, a}$ is an abuse of notation to refer to the elements of $u_2$ indexed by $a$, and analogously for $u_1$ and for $a^c$.*

Then, the next theorem shows how Von Kügelgen et al. (2021) can be adapted to prove that multi-view SSL methods that maximise the mutual information between the projections $\mathsf{I}(Z_1; Z_2)$ can obtain projections that capture and isolate the semantic information of the true latent variables.

**Theorem 5.** *Consider Assumption 1 and Assumption 2 and further assume that*

1. *the generative process $g$ is smooth, invertible and with a smooth inverse (i.e. a diffeomorphism);*

2. *$\mathsf{p}_{\tilde{Z}}$ is a smooth, continuous density on $\mathcal{Z}$ with $\mathsf{p}_{\tilde{Z}} > 0$ a.e., and*

3. *for any $j \in \{1, \ldots, n_u\}$, there is an $a \subseteq \{1, \ldots, n_u\}$ such that $j \in a$, $\mathsf{p}_A(a) > 0$, $\mathsf{p}_{U_{2, a} | U_{1, a} = u_{1, a}}(u_{2, a})$ is smooth with respect to both $u_{1, a}$ and $u_{2, a}$, and for any $u_{1, a}$ it holds that $\mathsf{p}_{U_{2, a} | U_{1, a} = u_{1, a}}(u_{2, a}) > 0$ for all $u_{2, a}$ in some open, non-empty subset containing $u_{1, a}$.*

*If the parameterised network function $\pi \circ f$ is smooth, the projections space is $(0, 1)^d \subseteq \mathbb{R}^d$, and the parameters $\theta$ are found to maximise the mutual information $\mathsf{I}(Z_1; Z_2)$ lower bound (1) with the reconstruction density $\mathsf{q}_{Z_2 | Z_1 = z_1}(z_2) = C_{\text{gauss}}(1)e^{-\|z_2 - z_2\|_2^2}$, then there is a bijection between the projections $Z$ and the true semantic variables $S$.*

*Proof.* The proof follows directly by Von Kügelgen et al. (2021, Theorem 4.4) by noting that the minimising the mutual information lower bound (1) with the reconstruction density $\mathsf{q}_{Z_2 | Z_1 = z_1}(z_2) = C_{\text{gauss}}(1)e^{-\|z_2 - z_2\|_2^2}$ coincides with the minimisation of their theorem. □

## C CONTRASTIVE LEARNING METHODS AND MUTUAL INFORMATION

As mentioned in the main text, contrastive learning methods are inspired by the contrastive loss in the InfoNCE Oord et al. (2018). The objective in Oord et al. (2018) is to maximise the mutual

information between the representations $R$ and the data $X$. To do so, they consider a batch of $k$ i.i.d. samples $X^{(1:k)}$ and try to ensure that the representation $R^{(i)}$ of sample $X^{(i)}$ can correctly predict which sample generated it among all other samples from the batch. More precisely, they minimise

$$\mathcal{L}_{\text{NCE}}(\theta) := -\frac{1}{k} \sum_{i=1}^{k} \mathbb{E}\left[ \log\left( \frac{s(X^{(i)}, R^{(i)})}{\sum_{j=1}^{k} s(X^{(i)}, R^{(j)})} \right) \right] \tag{9}$$

using gradient descent, where $s$ is some score or similarity function. Then, it is proven that minimising (9) maximises lower bound on the mutual information between the data and the representations, namely $\mathsf{I}(X; R) \geq \log k - \mathcal{L}_{\text{NCE}}$ (Poole et al., 2019).

### C.1 CONTRASTIVE MULTIVIEW CODING

Consider the `InfoNCE` loss and substitute $X$ by $Z_2$ and $R$ by $Z_1$ to maximise $\mathsf{I}(Z_1; Z_2)$. This results in

$$-\frac{1}{k} \sum_{i=1}^{k} \mathbb{E}\left[ \log\left( \frac{s(Z_2^{(i)}, Z_1^{(i)})}{\sum_{j=1}^{k} s(Z_2^{(i)}, Z_1^{(j)})} \right) \right],$$

and further considering the symmetric loss (i.e., the `InfoNCE` loss substituting $X$ by $Z_1$ and $R$ by $Z_2$) recovers $\mathcal{L}_{\text{CMC}}$. Hence, the `CMC` loss is exactly an `InfoNCE` loss with the temperature normalised cosine similarity as the score function, $s(\cdot, \cdot) = \exp\left(\text{sim}(\cdot, \cdot)/\tau\right)$, and therefore minimising it maximises a lower bound on $2\mathsf{I}(Z_1; Z_2)$.

For the analysis based on the mutual information decomposition (1), consider the re-writing of the $\mathcal{L}_{\text{CMC}}$ loss as follows

$$\mathcal{L}_{\text{CMC}}(\theta) = \sum_{b=1}^{2} \left( \frac{1}{k} \sum_{i=1}^{k} \text{sim}(Z_b^{(i)}, Z_{\bar{b}}^{(i)})/\tau - \frac{1}{k} \sum_{i=1}^{k} \log\left( \sum_{j=1}^{k} \exp\left(\text{sim}(Z_b^{(i)}, Z_{\bar{b}}^{(j)})/\tau\right) \right) \right). \tag{10}$$

Consider the term when $b = 2$ and focus inside the parenthesis, since the analysis is analogous when $b = 1$. In the reconstruction term from (1), letting the reconstruction density $\mathsf{q}_{Z_2 | Z_1 = z_1}$ be a von Mises–Fisher density with mean direction $z_1$ and parameter $1/\tau$, that is

$$\mathsf{q}_{Z_2 | Z_1 = z_1}(z_2) = C_{\text{vmf}}^{-1}(\tau) e^{\text{sim}(z_2, z_1)/\tau},$$

recovers the first term in the parenthesis up to additive constants that do not affect the optimisation.

For the second term, consider Joe (1989)'s KDE estimator with a standard Gaussian kernel density $\mathsf{q}$. Here, we may note that the restriction of the Gaussian density to elements of norm 1 is indeed a von Mises–Fisher density and that therefore have that for all $z, \tilde{z} \in \mathbb{S}^{d-1}$

$$\mathsf{q}\left( \frac{z - \tilde{z}}{\sqrt{\tau}} \right) = C_{\text{gauss}}^{-1}(1) e^{-\frac{\|z - \tilde{z}\|^2}{2\tau}} = C_{\text{gauss}}^{-1}(1) e^{-1/\tau} e^{\text{sim}(z, \tilde{z})/\tau}$$

when choosing a bandwidth $\sqrt{\tau}$.

Then, if $Z_1^{(i)}$ and $Z_2^{(j)}$ were to come from the same distribution, Joe (1989) estimator with the above kernel density would recover the second term up to additive constants that do not affect the optimisation. This would be the case if $2k$ augmentations' transformations were sampled randomly from a set $\mathcal{T}$ and applied independently to each data sample and 2 of them were applied independently to each data sample $X^{(i)}$ to generate the views $V_1^{(i)}, V_2^{(i)}$. However, this is often not the case and only 2 augmentations are sampled and applied to the whole batch of samples to generate the views (Chen et al., 2020a) or one branch is dedicated to generate only global views, while the other also generates local views (Caron et al., 2020).

### C.2 SIMCLR

#### C.2.1 SIMCLR AND THE INFONCE

Contrary to `CMC`, the `SimCLR` loss is not directly an `InfoNCE` loss, since there are terms in the denominator of the logarithm that are neither independent nor identically distributed. Indeed, $Z_1^{(j)}$

and $Z_2^{(j)}$ are generally not identically distributed due to differences in the transformations from each branch, and they are not independent since they are both obtained from $X^{(j)}$.

In fact, Poole et al. (2019)'s proof techniques cannot be directly modified to ensure that it is, even under the assumptions that the samples of each batch $X^{(1:k)}$ are i.i.d. and both the encoder $f_\theta$ and projector $\pi_\theta$ networks are deterministic and do not employ batch normalisation, which effectively ensures that $Z_1^{(1:k)}$ and $Z_2^{(1:k)}$ are i.i.d. and that $X^{(i)} \perp\!\!\!\perp Z_1^{(j)}$ and $X^{(i)} \perp\!\!\!\perp Z_2^{(j)}$ for all $i \neq j$. To see this, note that $\mathsf{I}(Z_1^{(1:k)}, Z_2^{(1)}; Z_2^{(1:k)}) = \mathsf{I}(Z_1^{(1)}; Z_2^{(1)})$ since $Z_1^{(1)}$ only depends on $Z_2^{(1)}$ via $X_1$ and is independent of all other projections. Then, similarly to (Poole et al., 2019), one may employ Donsker-Varadhan's variational bound on the relative entropy (Gray, 2011, Theorem 2.2.1) and see that

$$\mathsf{I}(Z_1^{(1:k)}, Z_2^{(1)}; Z_2^{(1:k)}) \geq \mathbb{E}\big[c(Z_1^{(1:k)}, Z_2^{(1:k)})\big] - \log \mathbb{E}\Big[\exp\big(c(Z_1^{(1:k)}, Z_2'^{(1)}, Z_2^{(1:k)})\big)\Big], \quad (11)$$

where $c$ is a critic function such that the above expectations are defined and $Z_2'^{(1)}$ is identically distributed to $Z_1^{(1)}$ but independent of all other random variables. Then, choosing the critic

$$c(z_1^{(1:k)}, z_2^{(1:k)}) = \log\left(\frac{s(z_2^{(1)}, z_1^{(1)})}{\sum_{b'=1}^2 \sum_{j=1}^k \mathbb{I}\big((1,2) \neq (j,b')\big) s(z_2^{(i)}, z_{b'}^{(j)})}\right),$$

ensures that the lower bound on $\mathsf{I}(Z_1^{(1:k)}, Z_2^{(1)}; Z_2^{(1:k)})$ is tight (an equality) if

$$\log\left(\mathsf{p}_{Z_1^{(1:k)}, Z_2^{(2:k)}|Z_2^{(1)}=z_2^{(1)}}(z_1^{(1:k)}, z_2^{(2:k)})\right) = c(z_1^{(1:k)}, z_2^{(1:k)}) + \phi(z_1^{(1:k)}, z_2^{(1:k)})$$

for any function $\phi$. Then, after averaging over all pairs $(Z_1^{(i)}, Z_2^{(j)})$, the first term in (11) becomes the SimCLR loss for a score function $s(\cdot, \cdot) = \exp\big(\mathrm{sim}(\cdot, \cdot)/\tau\big)$. However, the second term becomes

$$\frac{1}{2k}\sum_{b=1}^k \sum_{i=1}^k \log\left(\mathbb{E}\Big[\frac{s(Z_b'^{(i)}, Z_{\bar{b}}^{(i)})}{\sum_{b'=1}^2 \sum_{j=1}^k \mathbb{I}\big((i,b) \neq (j,b')\big) s(Z_b^{(i)}, Z_{b'}^{(j)})}\Big]\right),$$

where again $Z_b'^{(i)}$ is identically distributed to $Z_b^{(i)}$ but independent of all other random objects. Let us focus on the term $(i,b) = (1,2)$ as in the beginning of this note, that is

$$\log\left(\mathbb{E}\Big[\frac{s(Z_2'^{(1)}, Z_{\bar{b}}^{(i)})}{\sum_{b'=1}^2 \sum_{j=1}^k \mathbb{I}\big((1,2) \neq (j,b')\big) s(Z_2^{(1)}, Z_{b'}^{(j)})}\Big]\right).$$

To eliminate this term, one could use the fact that $Z_1^{(i)}$ are i.i.d. and take the average of these terms as in (Poole et al., 2019) in order to end up with the desired $\log k$ term. However, this is not possible since in the denominator there are additional terms associated with $Z_2^{(j)}$ for $j \neq 2$. Moreover, it is also not possible to include the terms $Z_2^{(j)}$ in this fictional average and end up with a term $\log(2k-1)$ since $Z_2^{(j)}$ and $Z_1^{(j)}$ are not i.i.d. as previously discussed.

However, consider again the assumption that the samples of each batch $X^{(1:k)}$ are i.i.d. and both the encoder $f_\theta$ and projector $\pi_\theta$ networks are deterministic and do not employ batch normalisation. Further consider the score function

$$s(x, z_1) = \begin{cases} \exp\big(\mathrm{sim}(\pi_\theta \circ f_\theta \circ t_2(x), z_1)/\tau\big) & \text{if } \pi_\theta \circ f_\theta \circ t_1(x) = z_1 \\ \sum_{b=1}^2 \exp\big(\mathrm{sim}(\pi_\theta \circ f_\theta \circ t_b(x), z_1)/\tau\big) & \text{otherwise} \end{cases},$$

where $t_1$ and $t_2$ are the *known* transformations of each branch. Then, as in the InfoNCE we observe that

$\mathsf{I}(X; Z_1|A = (t_1, t_2)) \geq$

$$\log k + \frac{1}{k}\sum_{i=1}^k \mathbb{E}\left[\log\left(\frac{\exp\big(\mathrm{sim}(Z_2^{(i)}, Z_1^{(i)})/\tau\big)}{\sum_{b'=1}^2 \sum_{j=1}^k \mathbb{I}\big((i,1) \neq (j,b')\big) \exp\big(\mathrm{sim}(Z_{b'}^{(j)}, Z_1^{(i)})/\tau\big)}\right)\bigg| A = (t_1, t_2)\right],$$

where $A$ denotes the augmentation random transformations. Taking the expectation in both sides with respect to the augmentations leads to the same inequality on $\mathsf{I}(X : Z_1|A)$. Then,

$$
\begin{aligned}
\mathsf{I}(X; Z_1|A) &= \mathsf{I}(X; Z_1, A) - \mathsf{I}(X; A) \\
&= \mathsf{I}(X; Z_1, A) \\
&= \mathsf{I}(X; Z_1) + \mathsf{I}(X; A|Z_1) = \mathsf{I}(X; Z_1)
\end{aligned}
$$

where the second equality holds since the augmentations are drawn independently from the data, and the last one holds from the Markov chain $X \to Z \leftarrow A$.

Note that here the assumptions are heavily employed: first, they are needed for the usual requirement in the `InfoNCE` that $\mathsf{I}(X^{(1:k)}; Z_1^{(i)}|A = (t_1, t_2)) = \mathsf{I}(X^{(i)}; Z_1^{(i)}|A = (t_1, t_2))$, and second, they are needed to perform the comparison $\pi_\theta \circ f_\theta \circ t_1(x) = z_1$. Taking the expectation with respect to the augmentations $A$ and averaging with the same observation on $\mathsf{I}(X; Z_2|A)$ reveals that

$$
\frac{1}{2}\big(\mathsf{I}(X; Z_1) + \mathsf{I}(X; Z_2)\big) \geq \log k - \mathcal{L}_{\text{SimCLR}}(\theta). \tag{12}
$$

From (12), we can see how minimising the `SimCLR` loss does not only minimises the information between the projections $\mathsf{I}(Z_1; Z_2)$, but also the information that each projection has about the data that is not shared by the other projection. Namely,

$$
\mathsf{I}(Z_1; Z_2) + \frac{1}{2}\big(\mathsf{I}(X; Z_1|Z_2) + \mathsf{I}(X; Z_2|Z_1)\big) \geq \log k - \mathcal{L}_{\text{SimCLR}}(\theta),
$$

where we used the equality $\mathsf{I}(X; Z_1) = \mathsf{I}(X, Z_2; Z_1) = \mathsf{I}(Z_1; Z_2) + \mathsf{I}(X; Z_1|Z_2)$ and its analogous for $\mathsf{I}(X; Z_2)$. Therefore, the above bound gives no guarantee that `SimCLR` learns projections that capture the semantic information, since $\mathsf{I}(X; Z_1|Z_2)$ could be made arbitrarily large capturing all the information not shared by the views. As shown in the approximate analysis with KDEs, this bound is crude.

### C.2.2  SIMCLR AND KDE

As with `CMC`, the `SimCLR` loss may be re-written for an analysis based on the mutual information decomposition from (1). Namely, $\mathcal{L}_{\text{SimCLR}}(\theta)$ is

$$
\frac{1}{2}\sum_{b=1}^{2}\left(\frac{1}{k}\sum_{i=1}^{k}\text{sim}(Z_b^{(i)}, Z_{\bar{b}}^{(i)})/\tau - \frac{1}{k}\sum_{i=1}^{k}\log\Big(\sum_{b'=1}^{2}\sum_{j=1}^{k}\mathbb{I}\big((i,b)\neq(j,b')\big)\exp\big(\text{sim}(Z_b^{(i)}, Z_{\bar{b}}^{(j)})/\tau\big)\Big)\right). \tag{13}
$$

Similarly to before, considering a von Mises-Fisher reconstruction density with parameter $1/\tau$ recovers the first term of the parenthesis for both $b = 1$ and $b = 2$. Also, considering the second term in the parenthesis individually for $b = 1$ and $b = 2$ introduces similar problems to those in `CMC` since $Z_1^{(i)}$ and $Z_2^{(i)}$ are not identically distributed and, even if they were, they are not independent.

Consider now the second term with the whole sum, i.e.

$$
\frac{1}{2k}\sum_{b=1}^{2}\sum_{i=1}^{k}\log\Big(\sum_{b'=1}^{2}\sum_{j=1}^{k}\mathbb{I}\big((i,b)\neq(j,b')\big)\exp\big(\text{sim}(Z_b^{(i)}, Z_{\bar{b}}^{(j)})/\tau\big)\Big),
$$

and model the samples as coming from the distribution $\mathbb{P}_Z = \mathbb{P}_{Z_1} + \mathbb{P}_{Z_2}$. Then, this term is approximately Joe (1989)'s estimator of $\mathsf{H}(Z)$, with the exemption that some pairs of the elements in the average are not independent of each other, which may harm its performance. However, the bias of this dependence is to balance the number of samples from each distribution of the mixture, and thus is not severe. Considering this estimator, disregarding the constants that do not affect the optimisation, and using the we obtain that

$$
\mathsf{I}(Z_1; Z_2) \gtrapprox \log k - \mathcal{L}_{\text{SimCLR}}(\theta) - \mathsf{D}_{\text{JS}}(\mathbb{P}_{Z_1}\|\mathbb{P}_{Z_2}).
$$

This equation also indicates that when both projections have similar marginals minimising the `SimCLR` loss approximately maximises the mutual information $\mathsf{I}(Z_1; Z_2)$.

C.3 CAVEATS OF THE ANALYSIS AND RELATED WORK.

**Caveats of the analysis.** Entropy estimation in high dimensions performs poorly: the rate of convergence decreases exponentially with the increasing dimensions (Joe, 1989). However, the highest entropy distribution on a finite, convex set $\mathcal{Z}$ is precisely the uniform distribution on that volume. Hence, these KDE estimators are good proxy's for the objective of maximising the entropy of the projections, since they force the samples to be maximally separated in $\mathcal{Z}$, and therefore are a good proxy for entropy maximisation and therefore mutual information maximisation. A similar observation is also done in (Wang & Isola, 2020; Zimmermann et al., 2021), where they assess that maximising the uniformity of the samples on the projections' space $\mathcal{Z}$ results in good performance. Moreover, these analyses require that the projections of different images on each branch are i.i.d., which is usually not the case due to the use of batch normalisation. The breaking of the i.i.d. assumption can be important in the `InfoNCE` lower bound on the mutual information, nonetheless, it does not discredit that the result of the KDE is a good proxy to maximise the entropy.

**Related work.** The following methods are variants of `CMC` or `SimCLR` and they inherit the analysis above either fully or partially. A whole new analysis is not written in the interest of space and repetition. In (Bachman et al., 2019), they adapt the Deep InfoMax (Hjelm et al., 2018, `DIM`) method to `CMC` and include the maximisation of local features and several level (layers of the network) features to the standard contrastive multi-view coding. In (Tian et al., 2020a), they consider `CMC` with multiple views instead of only two, and in (Tian et al., 2020b), they intend to learn the augmentations that best suit the information maximisation. Then, (Wu et al., 2018) and `MoCo` (He et al., 2020; Chen et al., 2020b) adopt the `CMC` setting without the symmetrisation (Figure 1c) and use additional negative pairs from the batches in previous iterations of training. Finally, in (Ramapuram et al., 2021), they adopt the `SimCLR` setting, where one of the two branches is modified to generate stochastic latent variables. More precisely, the representation $R_2 = f_\theta(X)$ is stochastically transformed to a latent variable $U \sim \mathbb{P}_{U|R}$ which is later decoded into the representation space $R_2' = \rho_\theta(U)$. The contrastive learning proceeds as usual with this reconstructed representation.

# D FURTHER PROJECTIONS' RECONSTRUCTION METHODS

In this section of the Appendix, we continue the analysis started in Section 3.2 on how different projections' reconstruction methods maximise the mutual information between the projections of different views.

**SimSiam** Chen & He (2021, `SimSiam`) consider a symmetric structure (Figure 1a) and, as `BYOL`, try to predict the projections from one branch using the predictions from the other branch and a small predictor network $g_\theta$. With this purpose, they try to minimise the negative cosine similarity

$$\mathcal{L}_{\text{SimSiam}}(\theta) := -\frac{1}{k} \sum_{b=1}^{2} \sum_{i=1}^{k} \text{sim}\big(g_\theta(Z_b), Z_{\bar{b}}\big),$$

which, as for `BYOL`, is equivalent to maximise the reconstruction term in (1) with a von Mises–Fisher reconstruction density after symmetrising with the reconstruction from the other branch.

Basically, `SimSiam` is `BYOL` in the extreme case where $\xi = \theta$. The realisation from Chen et al. (2020b) is that collapse can be avoided by making sure that in the optimisation of $\mathcal{L}_{\text{SimSiam}}(\theta)$ the parameters $\theta$ are only updated so that $g_\theta(Z_b)$ gets closer to $Z_{\bar{b}}$ and not so that $Z_{\bar{b}}$ also gets closer to $g_\theta(Z_b)$. They ensure that using a `stop-gradient` operator to $Z_{\bar{b}}$. However, this is sensitive to the employed parameters as seen in the experiments in `BYOL` when $\xi = \theta$ (Grill et al., 2020) and even in (Chen & He, 2021), where they see that the method collapses or is unstable without a predictor network and depending on where in the network batch normalisation is employed.

Similarly, `SimSiam` also considers a variant with a cross-entropy loss. Practically, they try to make sure that the predictions from one branch generate a similar distribution on $[d]$ as the projections from the other branch. That is, they try to minimise the cross entropy

$$\mathcal{L}_{\text{SimSiam}}(\theta) := \frac{1}{k} \sum_{b=1}^{2} \sum_{i=1}^{k} \mathsf{s}\big(g_\theta(Z_b)\big)^\intercal \log \mathsf{s}\big(Z_{\bar{b}}\big),$$

where $s$ is the `softmax` function. Theoretically, this amounts to generating a surrogate variable $W_b = \phi(Z_b)$ and trying to maximise $I(Z_1; W_2) + I(W_1; Z_2)$, thus maximising $2I(Z_1; Z_2)$ due to the data processing inequality (c.f. Figure 1b). Focus on $b = 2$ and the decomposition of $I(Z_2; W_2)$ using (1). Then, letting the reconstruction density be $q_{W_2|Z_1=z_1} = s\big(g_\theta(z_1)\big)$ recovers $\mathcal{L}_{\text{SimSiam}}(\theta)$. Finally, the same analysis as before holds for the study entropy terms $H(Z_1)$ and $H(Z_2)$ (or collapse prevention).

**DeepCluster and SWAV** Both (Caron et al., 2018; Asano et al., 2019, `DeepCluster`) and (Caron et al., 2020, `SwAV`) generate discrete surrogate variables $W_b = \phi(Z_b)$ based on a clustering approach and try to minimise a cross-entropy term similar to that in $\mathcal{L}_{\text{SimSiam}}$. Then, they ensure that the entropy is large by engineering the clustering so that it is balanced (i.e., there are enough projections assigned to each cluster). For the rest of the section let $\mathcal{Z} \subseteq \mathbb{R}^d$ and $\mathcal{W} = [m]$.

`DeepCluster` has an asymmetric setting (Figure 1d) but with $\xi = \theta$. First, the cluster assignments $W_2^{(i)} = \phi(Z_2^{(i)})$ of all the data points are obtained solving the problem[5]

$$C^\star = \operatorname*{arg\,inf}_{C \in \mathbb{R}^{d \times m}} \frac{1}{n} \sum_{i=1}^{n} \|Z_2^{(i)} - C\mathsf{p}_2^{(i)}\|^2 \quad \text{s.t.} \quad \|\mathsf{p}_2^{(i)}\|_\infty = 1, \ \mathsf{p}_2^{(i)} \in \{0, 1\}^m,$$

where $C^\star$ represent the $m$ centroids of the clusters in $\mathcal{Z}$ and $\mathsf{p}_2^{(i)}$ is the p.m.f. of $W_2^{(i)}$ given $Z_2^{(i)}$. Then, the parameters $\theta$ are optimised by minimising the cross entropy

$$\mathcal{L}_{\text{DeepCluster}}(\theta) := \frac{1}{k} \sum_{i=1}^{k} \left(\mathsf{p}_2^{(i)}\right)^\mathsf{T} \log s\big(g_\theta(Z_1^{(i)})\big),$$

where $g_\theta : \mathcal{Z} \to \mathbb{R}^m$ is a small predictor network and clearly $q_{W_2|Z_1=z_1} = s \circ g_\theta(z_1)$. This optimisation amounts to maximising the reconstruction term in (1) for $I(Z_1; W_2) \le I(Z_1; Z_2)$. However, without accounting for the entropy term $H(W_2)$ the loss $\mathcal{L}_{\text{DeepCluster}}$ is trivially minimised by assigning all projections to a single cluster and always predicting such a cluster (i.e. $H(W_2) = 0 \implies I(Z_2; W_2) = 0$). Caron et al. (2020) circumvent this issue by randomly reassigning points from full clusters to empty clusters and sampling the images of each batch based on a uniform distribution on the clusters' labels. Hence, effectively keeping a high entropy $H(W_2)$.

`SwAV` has a symmetric setting (Figure 1b). Let us focus on one branch ($b = 2$), since the analysis of the method then follows by symmetrisation. Here, the cluster assignments $W_2^{(i)} = \phi(Z_2^{(i)})$ are obtained solving the following optimisation problem

$$P_2 = \operatorname*{arg\,max}_{P \in \mathcal{P}} \left\{ \operatorname{Tr}\left(Z_2^{(1:k)} C^\mathsf{T} P^\mathsf{T}\right) + \epsilon H(P) \right\}$$

using the Sinkhorn-Knopp algorithm (Sinkhorn, 1974; Cuturi, 2013), where $Z_2^{(1:k)} \in \mathbb{R}^{k \times d}$, $C \in \mathbb{R}^{m \times d}$ are the $m$ centroids (or prototypes) in $\mathbb{R}^d$, $\mathcal{P} = \{P \in \mathbb{R}_+^{k \times m} : P^\mathsf{T} \mathbf{1}_k = \mathbf{1}_m/m \text{ and } P\mathbf{1}_m = \mathbf{1}_k/k\}$ is the transportation polytope, and $\mathbf{1}_k$ is the all ones vector in $\mathbb{R}^k$. Let $C^{(i)}$ and $P_2^{(i)}$ denote the $i$-th row of $C$ and $P_2$, respectively. In `SwAV`, both the projections and the prototypes lay in the unit hypersphere, i.e. $Z^{(i)}, C^{(i)} \in \mathbb{S}^{d-1}$, and as seen in `BYOL` maximising the dot product is equivalent to minimising the squared $\ell_2$ norm distance. Hence, the optimisation problem above is equivalent to the otpimal transport problem of moving the $k$ samples $Z_2^{(1:k)} \in \mathbb{R}^d$ to the positions of the $m$ prototypes $C$ with the minimum $\ell_2$ distance cost. Moreover, to aid the optimisation calculations, an entropic regularisation is included and solved using the Sinkhorn-Knopp algorithm (Sinkhorn, 1974; Cuturi, 2013)[6], where $H(P_2) := \sum_{i=1}^{k} \left(P_2^{(i)}\right)^\mathsf{T} \log P_2^{(i)}$.

Note that the $j$-th element of $P_2^{(i)}$ can be understood as the probability of assigning $Z_2^{(i)}$ to the cluster $W_2^{(i)} = j$. The optimisation aims to have $P_2 \in \mathcal{P}$ and therefore $P_2^\mathsf{T} \mathbf{1}_k \approx \mathbf{1}_m/m$, which by this interpretation would mean that $\mathsf{p}_{W_2} \approx \mathbf{1}_m/m$, thus maximising the desired entropy $H(W_2)$

---

[5]In (Asano et al., 2019), the clusters are obtained solving an optimal transport problem similar to `SwAV`.

[6]Actually, `SwAV` only *approximately* solves the problem, since the algorithm is run for only three steps.

in (1). Then similarly to `DeepCluster`, the reconstruction term in (1) for $\mathsf{I}(Z_1; W_2)$ is maximised minimising the `SwAV` loss

$$\mathcal{L}_{\text{SwAV}}(\theta) := \frac{1}{k} \sum_{b=1}^{2} \sum_{i=1}^{k} \left(\mathsf{p}_2^{(i)}\right)^{\mathsf{T}} \log \mathsf{s}\left(C Z_{\bar{b}}^{(i)}\right),$$

for $b = 2$, where $\mathsf{p}_2^{(i)} = P_2^{(i)}/(\mathbf{1}_m^{\mathsf{T}} P_2^{(i)})$ and $\mathsf{q}_{W_2|Z_1=z_1} = \mathsf{s}(Cz_1)$, hence maximising the mutual information $\mathsf{I}(Z_1; W_2)$. An analogous analysis for the branch $b = 1$ reveals that minimising $\mathcal{L}_{\text{SwAV}}$ with the entropic regularisation assignment maximises the mutual information $\mathsf{I}(Z_2; W_1)$. In `SwAV` (and hence in this analysis), the prototypes are treated as parameters of the network (i.e., $C \in \theta$) and are updated using stochastic gradient descent to minimise $\mathcal{L}_{\text{SwAV}}$.

## E    ENTREC ALGORITHM

Below, we sketch `EntRec`'s main algorithm.

---

**Algorithm 1** `EntRec`'s main learning algorithm.

---

**Input:** Dataset $\mathcal{D} = \{x^{(i)}\}_{i=1}^{n}$, batch size $k$, reconstruction density $\mathsf{q}_{\text{Rec}}$, kernel density $\mathsf{q}_{\text{KDE}}$, encoder and projector networks $f_\theta$ and $\pi_\theta$, augmentation set $\mathcal{T}$, and number of iterations `iterations`.

1: Set `iteration` $= 1$
2: **while** `iteration` $\leq$ `iterations` **do**
3:     Draw a batch $x^{(1:k)}$ uniformly at random from the dataset $\mathcal{D}$.
4:     Draw two augmentation functions $t_1$ and $_2$ uniformly at random from $\mathcal{T}$.
5:     **for all** $i \in \{1, \ldots, k\}$ **do**
6:         Calculate $z_1^{(i)} = \pi_\theta \circ f_\theta \circ t_1(x^{(i)})$ and $z_2^{(i)} = \pi_\theta \circ f_\theta \circ t_2(x^{(i)})$.
7:     **end for**
8:     **if** `EntRecDisc` **then**
9:         Calculate $\mathsf{p}_1 = \frac{1}{k} \sum_{i=1}^{k} \mathsf{s}(z_1^{(i)})$ and $\mathsf{p}_2 = \frac{1}{k} \sum_{i=1}^{k} \mathsf{s}(z_2^{(i)})$.
10:     **end if**
11:     **for all** $i \in \{1, \ldots, k\}$ **do**
12:         **if** `EntRecCont` **then**
13:             **for all** $j \in \{1, \ldots, k\}$ **do**
14:                 Calculate $\ell_{\text{Ent},1}^{(i,j)}(\theta) = \mathsf{q}_{\text{KDE}}\left(\frac{z_1^{(i)} - z_2^{(j)}}{h}\right)$ and $\ell_{\text{Ent},2}^{(i,j)}(\theta) = \mathsf{q}_{\text{KDE}}\left(\frac{z_2^{(i)} - z_1^{(j)}}{h}\right)$.
15:             **end for**
16:             Calculate $\ell_{\text{Ent}}^{(i)}(\theta) = -\frac{1}{2}\left(\log\left(\frac{1}{kh^d} \sum_{j=1}^{k} \ell_{\text{Ent},1}^{(i,j)}\right) + \log\left(\frac{1}{kh^d} \sum_{j=1}^{k} \ell_{\text{Ent},2}^{(i,j)}\right)\right)$.
17:             Calculate $\ell_{\text{Rec}}^{(i)}(\theta) = \frac{1}{2}\left(\log\left(\mathsf{q}_{\text{Rec}}(z_2^{(i)}|z_1^{(i)})\right) + \log\left(\mathsf{q}_{\text{Rec}}(z_1^{(i)}|z_2^{(i)})\right)\right)$.
18:         **else if** `EntRecDisc`
19:             Calculate $\ell_{\text{Ent}}^{(i)}(\theta) = -\frac{1}{2}\left(\mathsf{p}_1^{\mathsf{T}} \log\left(\mathsf{p}_1\right) + \mathsf{p}_2^{\mathsf{T}} \log\left(\mathsf{p}_2\right)\right)$.
20:             Calculate $\ell_{\text{Rec}}^{(i)}(\theta) = \frac{1}{2}\left(\mathsf{s}(z_1^{(i)})^{\mathsf{T}} \log\left(\mathsf{s}(z_2^{(i)})\right) + \mathsf{s}(z_2^{(i)})^{\mathsf{T}} \log\left(\mathsf{s}(z_1^{(i)})\right)\right)$.
21:         **end if**
22:     **end for**
23:     Calculate $\mathcal{L}_{\text{EntRec}}(\theta) = -\frac{1}{k} \sum_{i=1}^{k} \left(\ell_{\text{Ent}}^{(i)}(\theta) + \ell_{\text{Rec}}^{(i)}(\theta)\right)$.
24:     Update encoder $f_\theta$ and projector $\pi_\theta$ to minimise $\mathcal{L}_{\text{EntRec}}(\theta)$.
25: **end while**
26: **return** encoder network $f_\theta$, and throw away $\pi_\theta$.

---

## F    ENTREC CONVERGENCE

In **Theorem 2** and **Theorem 3** we state the bias and variance properties of the `EntRec` estimators, which in turn describe their convergence in mean squared error (MSE). In this section of the Appendix, we formalise and complement these statements.

### F.1 ENTRECCONT CONVERGENCE

#### F.1.1 ENTROPY ESTIMATION AND SELECTION OF THE BANDWIDTH PARAMETER

The bias and variance of Joe (1989)'s KDE estimator $\hat{\mathsf{H}}_{\text{KDE}}$ of the entropy $\mathsf{H}(Z_b)$ are (Joe, 1989, Section 4, page 695)

$$\mathbb{B}[\hat{\mathsf{H}}_{\text{KDE}}] \in \mathcal{O}(k^{-1}h^{4-d}) + \mathcal{O}(k^{-2}h^{-2d}) + \mathcal{O}(h^4) \text{ and}$$

$$\mathbb{V}[\hat{\mathsf{H}}_{\text{KDE}}] \in \mathcal{O}(k^{-1}) + \mathcal{O}(k^{-2}h^{8-d}) + \mathcal{O}(k^{-2}h^{-d}) + \mathcal{O}(k^{-1}h^{8-d}) + \mathcal{O}(k^{-2}h^{4-2d}) + \mathcal{O}(h^8).$$

Hence, as long as $h \in \mathcal{O}(k^{-1/(d+\varepsilon)})$ for some small $\varepsilon > 0$ both the bias and the variance vanish, and the estimator convergences in MSE, even if it does so at a slow rate. Then, a sensible choice of the bandwidth is $h \approx 1$ since $k^{-1/(d+\varepsilon)} \to 1$ as $d$ increases.

Under the further assumption that the distribution of $Z_b$ is $\beta$-smooth (i.e., it belongs to the Hölder or Sobolev classes) then the bias and variance of the estimator are (Krishnamurthy & Wang, 2015)

$$\mathbb{B}[\hat{\mathsf{H}}_{\text{KDE}}] \in \mathcal{O}(h^\beta) \text{ and}$$

$$\mathbb{V}[\hat{\mathsf{H}}_{\text{KDE}}] \in \mathcal{O}(k^{-1}h^{-d}).$$

As previously, the bias and the variance of the estimator only vanish if $h \in \mathcal{O}(k^{-1/(d+\varepsilon)})$ for some small $\varepsilon > 0$, with the optimal choice $h = k^{-1/(d+2\beta)}$. Nonetheless, having a bias term independent of the parameter of the optimisation is not harmful in itself. Hence, when the KDE estimator is employed only for optimisation purposes both $h \in \mathcal{O}(k^{-1/(d+\varepsilon)})$ and $h \in \mathcal{O}(1)$ may work. For instance, for the experiments using the von Mises–Fisher distribution we set $h = 0.1$ to match the temperature employed by (Tian et al., 2020a, `CMC`) and (Chen et al., 2020a, `SimCLR`).

#### F.1.2 CROSS ENTROPY ESTIMATION

Note that $\log \mathsf{q}_{Z_b|Z_{\bar{b}}^{(i)}}(Z_b^{(i)})$ are independent and identically distributed random variables with expectation $\mathbb{E}[\log \mathsf{q}_{Z_b|Z_{\bar{b}}}(Z_b)]$. Hence, the empirical estimator is unbiased. Similarly, the variance of the estimator is $\mathbb{V}[\frac{1}{k}\sum_{i=1}^k \log \mathsf{q}_{Z_b|Z_{\bar{b}}^{(i)}}(Z_b^{(i)})] = \sigma_{\mathsf{q}}^2/k$, where $\sigma_{\mathsf{q}}^2 = \mathbb{V}[\log \mathsf{q}_{Z_b|Z_{\bar{b}}}(Z_b)]$.

Consider now that a reconstruction density is of the form $\mathsf{q}_{Z_b|Z_{\bar{b}}=z_{\bar{b}}}(z_b) = Ce^{-\rho(z_b,z_{\bar{b}})}$ and that the projections lay in a convex body $\mathcal{Z} \in \mathbb{R}^d$. Then, we know that $\log \mathsf{q}_{Z_b|Z_{\bar{b}}=z_{\bar{b}}}(z_b) \in [\log C - \rho(\mathcal{Z}), \log C]$, where $\rho(\mathcal{Z})$ is the diameter of $\mathcal{Z}$ with respect to $\rho$. Therefore, by the Popoviciu's inequality on variances we have that $\sigma_{\mathsf{q}}^2 \le \rho(\mathcal{Z})^2/4$, which implies that for $\rho(\mathcal{Z}) < \infty$ the estimator converges in MSE. This holds for the two cases considered in this paper:

- Von Mises–Fisher distribution in $\mathcal{Z} = \mathbb{S}^{d-1}$: Here the diameter with respect to $\rho(z_1, z_2) = \kappa \text{sim}(z_1, z_2)$ is $\rho(\mathcal{Z}) = \kappa^2$ and hence the estimator converges in MSE at a $\kappa^2/(4k)$ rate.

- Gaussian distribution in $\mathcal{Z} = [-1, 1]^d$: Here the diameter with respect to $\rho(z_1, z_2) = \|z_1 - z_2\|^2/(2\sigma^2)$ is $\rho(\mathcal{Z}) = 2d/\sigma^2$ and hence the estimator converges in MSE at a $d/(2k\sigma^2)$ rate.

### F.2 ENTRECDISC CONVERGENCE

#### F.2.1 ENTROPY ESTIMATION

The plug-in estimator $\hat{\mathsf{H}}_{\text{plug-in}}$ of the entropy $\mathsf{H}(W_b)$ is known to have the following bias and variance terms (see e.g. (Girsanov, 1959, Equations (3) and (4))) or (Antos & Kontoyiannis, 2001, Introduction)):

$$\mathbb{B}[\hat{\mathsf{H}}_{\text{plug-in}}] \in \mathcal{O}\left(\frac{d-1}{2k}\right) + \mathcal{O}\left(\frac{1}{k^2}\right) \text{ and}$$

$$\mathbb{V}[\hat{\mathsf{H}}_{\text{plug-in}}] \in \mathcal{O}\left(\frac{\sigma_{\mathsf{p}}^2}{k}\right) + \mathcal{O}\left(\frac{1}{k^2}\right),$$

where $\sigma_{\mathsf{p}}^2 = \mathbb{V}[-\log \mathsf{p}_{W_b}(W_b)]$. The bias and the variance vanish as long as $d$ is fixed and $\sigma_{\mathsf{p}}^2 < \infty$, meaning that the estimator converges in MSE.

Note that $\mathsf{p}_{W_b} = \mathbb{E}[\mathsf{s}(Z_b)]$, where $\mathsf{s}$ is the softmax operator. Hence, we have that

$$\mathbb{V}[-\log \mathsf{p}_{W_b}(W_b)] \leq \mathbb{E}[\log^2 \mathsf{p}_{W_b}(W_b)]$$

$$\leq \mathbb{E}\left[\left(Z_{b,W_b} - \log\left(\sum_{i=1}^{k} e^{Z_{b,i}}\right)\right)^2\right]$$

$$\leq \mathbb{E}\left[(\log d + Z_{b,\max} - Z_{b,\min})^2\right],$$

where the first inequality follows from the fact that $\mathbb{V}[X] \leq \mathbb{E}[X^2]$; the second from Jensen's inequality and the formula of the softmax; and the last one from the log-sum-exp trick. Here, $Z_{b,i}$ denotes the $i$-th element of the random vector $Z_b$.

In the particular case where the projections lie in the sphere $\mathbb{S}^{d-1}$ we have that $\sigma_p^2 \leq (\log d + 1)^2$. Similarly, if they lay in the cube $[-1,1]^d$, we have that $\sigma_p^2 \leq (\log d + 2)^2$. Therefore, under these standard conditions the variance vanishes at a rate in $\mathcal{O}(\log^2(d)/k) + \mathcal{O}(1/k^2)$.

### F.2.2 CROSS ENTROPY ESTIMATION

As in Appendix **F.1.2**, note that $\log \mathsf{q}_{W_b|Z_{\bar{b}}^{(i)}}(W_b^{(i)})$ are independent and identically distributed random variables with expectation $\mathbb{E}[\log \mathsf{q}_{W_b|Z_{\bar{b}}}(W_b)]$. Hence, the empirical estimator is unbiased. Similarly, the variance of the estimator is $\mathbb{V}[\frac{1}{k}\sum_{i=1}^{k} \log \mathsf{q}_{W_b|Z_{\bar{b}}^{(i)}}(W_b^{(i)})] = \sigma_{\mathsf{q}}^2/k$, where $\sigma_{\mathsf{q}}^2 = \mathbb{V}[\log \mathsf{q}_{W_b|Z_{\bar{b}}}(W_b)]$. Hence, the variance vanishes as long as $\sigma_{\mathsf{q}}^2 < \infty$, meaning that the estimator converges in MSE.

As for the entropy estimation, note that $\mathsf{q}_{W_b|Z_{\bar{b}}}(W_b) = \mathsf{s}(Z_{\bar{b}})$. Hence, repeating the analysis above in Appendix **F.2.1** we obtain that $\sigma_{\mathsf{q}}^2 \leq \mathbb{E}\left[(\log d + Z_{\bar{b},\max} - Z_{\bar{b},\min})^2\right]$ and therefore for projections in the sphere $\mathbb{S}^{d-1}$ or the cube $[-1,1]^d$ the variance vanishes at a rate in $\mathcal{O}(\log^2(d)/k) + \mathcal{O}(1/k^2)$.

## G EXPERIMENT PROTOCOL DETAILS

### G.1 PRE-TRAINING

We do unsupervised pre-training on the 1000-class ImageNet training set (Deng et al., 2009) without using labels. Unless specified, our explorations use the following settings for unsupervised pre-training:[7]

- *Optimizer.* We use LARS for pre-training (You et al., 2017) and apply a scaling of lr $\times$ Batchsize / 256 (linear scaling (Goyal et al., 2017)). The base lr = 0.1, except for `BYOL` and `DINO` (lr = 0.3). The learning rate has a cosine decay schedule (Loshchilov & Hutter, 2016; Chen et al., 2020a) with an initial linear warm-up for 10 epochs. The weight decay is 1e-4, except for `BYOL` and `DINO` (1e-6). We use batch normalization (BN) (Ioffe & Szegedy, 2015) synchronized across devices, following Chen et al. (2020a).

- *Backbone.* We use ResNet-50 (He et al., 2016) as the default backbone.

- *Projection MLP.* The projection MLP has 3 layers. The hidden fc is 2048-d, except for `BYOL` when it is 4096-d. The output size is 128-d, except for `BYOL` (256-d) and `DINO` (60000-d).

- *Temperature.* The temperature $\tau$ for `SimCLR` and `CMC` is 0.1. For `DINO`, the student temperature is 0.1, while the teacher temperature is 0.07.

- *Augmentations.* We use `DINO` augmentations (Caron et al., 2021) for all methods with 2 global crops (cropping scale [0.4, 1.0]) and 8 local crops (cropping scale [0.05, 0.4]).

---

[7] These hyperparameters have been optimised by us to obtain the best performance of each of the SSL methods with our implementation. For `EntRec`, we just used the parameters from `SimCLR` without any further optimisation.

- *EntRecCont reconstruction density.* In Section **4.1** we intentionally left the reconstruction density $q_{Z_b|Z_{\bar{b}}=Z_{\bar{b}}^{(i)}}$ of EntRecCont unspecified to highlight the generality of our theoretical results. In the experiments, we choose this to be a von Mises–Fisher density with mean direction $Z_{\bar{b}}^{(i)}$ and parameter 10, so that they match the one used by SimCLR and CMC. Nonetheless, in Appendix **H.3** we give the results of EntRecCont with a Gaussian distribution with mean $Z_{\bar{b}}^{(i)}$ and variance 0.99,[8][9] where we can see they are of the same order than those obtained with the von Mises–Fisher distribution.

- *Float precision.* We use half precision for all methods and experiments, except for DINO where full precision was found to be needed.

## G.2 EVALUATION

On ImageNet, the evaluation of the learned representations is done by training a supervised linear classifier on frozen representations for one epoch in the training set, and then testing it in the test set. For transfer learning, the evaluation of the learned representations is done by supervised fine-tuning of the learned representation together with a linear classifier on the respective training set for 200000 steps with a batch size of 256, following Chen et al. (2020a). We fine-tune using the RM-SProp (Hinton et al., 2012) with a learning rate of 0.0003 and a cosine decay schedule with initial linear warm-up for 5 epochs. As augmentations during fine-tuning, we use standard ImageNet augmentations (random cropping followed by random horizontal flips and normalisation, Krizhevsky et al. (2017). Testing is done on the respective test sets.

## H ADDITIONAL EXPERIMENTS

### H.1 TRANSFER LEARNING

Below analyse the transfer learning performance on a series of transfer tasks. Results can be found in Table **3**. Again, we see a performance of EntRec that is comparable with the performance of other multi-view SSL methods. Note that BYOL and DINO results were not finished at the time of submission, but will be added later.

Table 3: Fine-tuning transfer learning results. Pre-training on ImageNet for 400 epochs with batch size 4096. Numbers are top-1 accuracy.

| Accuracy % | Flowers | CIFAR10 | CIFAR100 | Caltech-101 | Aircraft | Cars | Food | SUN397 | DTD | Mean |
|---|---|---|---|---|---|---|---|---|---|---|
| SimCLR | **85.2** | 96.1 | **82.1** | 78.7 | 67.3 | 87.1 | **87.1** | 54.1 | 57.6 | **77.2** |
| CMC | 82.2 | 96.0 | 81.9 | **79.5** | **68.5** | **88.4** | 86.6 | 54.0 | 57.6 | **77.2** |
| EntRecCont | 81.8 | **96.4** | 80.7 | 77.1 | 68.3 | 88.2 | 86.3 | 54.4 | 57.1 | 76.7 |
| EntRecDisc | 81.5 | 95.0 | 82.0 | **79.5** | 66.4 | 88.2 | 86.9 | **54.7** | **57.7** | 76.9 |

### H.2 THE EFFECT OF DIRECT ENTROPY MAXIMISATION ON ENTROPY

For the two methods that use discrete surrogate random variables (DINO, EntRecDisc), we can get accurate finite samples estimations of the entropy term in (1). This allows us to analyse how including this term into the training objective with EntRecDisc affects the behaviour of this term during training in comparison to what happens when engineering techniques are used to maintain high entropy. Figure **2** confirms that this indeed helps to maintain higher entropy: EntRecDisc maintains a non-decreasing level of entropy, while DINO slowly lowers the entropy term, however not totally collapsing.

For the methods that do not employ discrete surrogate variables, we cannot quantify how much of the potential entropy they capture, but we can observe qualitatively how this entropy is controlled during training through a KDE proxy (see Figure **3**). We see that for BYOL, it is dropping at some point, but then stabilising and not collapsing either. We think it would be interesting to explore the

---

[8]This variance is selected to comply with the kernel bandwidth requirements of **Theorem 2**.
[9]To fulfil the theoretical requirements of **Theorem 4**, we restrict the projections to the box $\mathcal{Z} = [-1, 1]^d$.

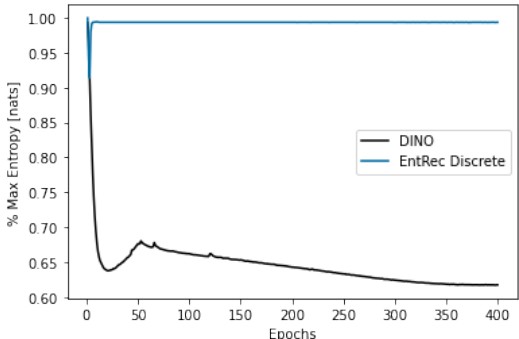

Figure 2: Entropy dynamics during training for `DINO` and `EntRecDisc`, which allow for accurate entropy estimation through their discrete probability distributions. The entropy is shown as a percentage of the total potential entropy $\log(|\mathcal{W}|)$.

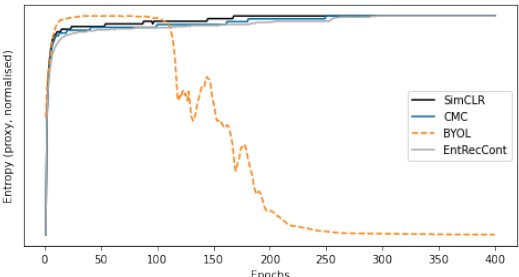

Figure 3: Entropy dynamics during training for `SimCLR`, `CMC`, `BYOL`, and `EntRecCont`. The y-axis is a proxy of the entropy obtained with a KDE, and has been normalised for each method based on the range obtained during the trajectory in order to have them all in one single plot. Note that this means that the curves can only be interpreted qualitatively and we are interested in the trend. For example, `BYOL` does not collapse to 0 entropy but simply stabilises at a level of entropy that is lower than where it started from.

performance levels `BYOL` could reach if a more direct way of maintaining or maximising the entropy term can be found for this method.

### H.3 RESULTS ON OTHER ENTREC RECONSTRUCTION DENSITIES

As mentioned above, we used a von Mises-Fisher density for the reconstruction density of `EntRecCont` in the experiments in the main part of this paper. However, `EntRecCont` and its theoretical properties are more general than this special case, and thus below we also include the results obtained for varying batch sizes and epochs on ImageNet with a Gaussian reconstruction density below in Table 4 and Table 5. We see that the results are still comparable to the remaining methods and to `EntRecCont` with a von Mises-Fisher reconstruction density (compare to Table 1 and Table 2).

Table 4: Performance on the ImageNet test set of `EntRecCont` with Gaussian reconstruction density across batch sizes after 400 epochs of training.

|  | Accuracy | $\Delta$Accuracy wrt. 4096 | | |
|---|---|---|---|---|
|  | 4096 | 2048 | 1024 | 512 |
| `EntRecCont` (Gauss) | 69.4 | -1.2 | -2.5 | -4.7 |

Table 5: Performance on the ImageNet test set of `EntRecCont` with Gaussian reconstruction density across epochs with batch size of 4096.

|  | Accuracy | $\Delta$Accuracy wrt. 400 | | |
|---|---|---|---|---|
|  | 400 | 300 | 200 | 100 |
| `EntRecCont` (Gauss) | 69.4 | -0.8 | -2.5 | -5.6 |

