# OpenReview forum: "On Information Maximisation in Multi-View Self-Supervised Learning"
_ICLR.cc/2023/Conference — Submitted to ICLR 2023_

### Official Review · Reviewer_KfpZ · 2022-10-19

**Confidence:** 4
**Correctness:** 2
**Technical Novelty And Significance:** 2
**Empirical Novelty And Significance:** 2
**Recommendation:** 3

**Clarity, Quality, Novelty And Reproducibility:**

The quality is fair, but can be improved. The contributions should be stated more transparently and the empirical results could be more thorough to warrant the claims made in the abstract. The paper is mostly clear and easy to follow.

The novelty is limited and, in my opinion, insufficient (see Weaknesses).


**Strength And Weaknesses:**

**Strengths:**
- The authors test two different estimators for the entropy terms, a KDE estimator and a plug-in estimator. It's also nice that the authors add information about the asymptotic behavior of these estimators, even if these results are not entirely novel.
- The authors reference critiques of the information-theoretic perspective on SSL objectives.



**Weaknesses**
- It is well known that many existing SSL methods can be connected via MI maximization and the contribution of the paper in this regard is not clear enough. For which methods is the MI maximization perspective actually new? This should be clearly stated in the abstract and/or introduction.
- Analogously, the decomposition of the MI into cross-entropy and entropy terms is well known and should not be stated as a contribution ("we show ...") in the abstract.
- The claim that the proposed objective "exactly optimises both the reconstruction and entropy terms" is misleading, because the model uses estimators.
- The "unification of current theoretical properties" is overstated. The two referenced papers (Zimmermann et al.; Kügelgen et al.) are quite similar in that they show identifiability results for multi-view nonlinear ICA and related settings. Specifically, Kügelgen et al. already discuss the connection of their work to the work from Zimmermann et al.
- The claimed "improved robustness" of one of the proposed models is not sufficiently clear from the experiments.

**Summary Of The Paper:**

The paper deals with the topic of mutual information (MI) maximization in multi-view self-supervised learning (SSL). The paper takes an information-theoretic perspective and shows that many current self-supervised learning methods maximize a lower bound on the MI between the representations of different views---a result that is not entirely novel. Based on a well-known decomposition of the MI into a cross-entropy and an entropy term, the paper argues that it can be beneficial to estimate these two terms individually. On the theoretical side, the authors draw a connection to two recent identifiability results related to nonlinear ICA. Empirically, the paper shows that models that estimate the individual (cross-)entropy terms perform on par with existing self-supervised learning methods.

**Summary Of The Review:**

The decomposition of the MI and estimation of the two individual terms (cross-entropy and entropy) is an interesting research direction, which can lead to new insights into SSL. However, in its current form, the paper provides insufficient contribution compared to the results from previous work. In particular, the authors should state their contribution more clearly and verify the benefits of the proposed objective more thoroughly.

---

> ### Author Response · Authors · 2022-11-17
> **Answer to Reviewer KfpZ [1/1]**
>
> We would like to thank the reviewer for their comments, which allowed us to clarify some parts of the paper. Please, find below our answers to them.
>
> * *It is well known that many existing SSL methods can be connected via MI maximization and the contribution of the paper in this regard is not clear enough. For which methods is the MI maximization perspective actually new? This should be clearly stated in the abstract and/or introduction.*
>
> Thank you for pointing out that this was not clear.
>
> As far as we know, the connection between maximisation of the mutual information and SSL algorithms was only known for InfoNCE-like methods (i.e., CMC and MoCo). We complement this knowledge with SimCLR (closing the picture of contrastive methods), BYOL, SimSiam, DINO, DeepCluster, and SwAV.
>
> In the reviewed version of the paper, we pointed this out explicitly as we agree with this reviewer that from the previous list of contributions this was not clear.
>
> * *Analogously, the decomposition of the MI into cross-entropy and entropy terms is well known and should not be stated as a contribution ("we show ...") in the abstract.*
>
> We acknowledge that the decomposition is known and we already cited a couple of references that used it before. Note that this decomposition was (and is) not listed in the bullet point list of contributions in section 1. The wording of the abstract was indeed unfortunate and we agree that could lead to misunderstandings so we followed the reviewer's suggestion and we have modified it as follows: <<Further, we **observe** that this bound decomposes into a "reconstruction" term, treated identically by all SSL methods, and an "entropy" term, where existing SSL methods differ in their treatment.>>
>
> * *The claim that the proposed objective "exactly optimises both the reconstruction and entropy terms" is misleading, because the model uses estimators.*
>
> We also agree that “exactly” should be changed to “directly” as we did throughout the rest of the paper. Thank you for pointing this out. We have updated it in the reviewed version.
>
> * *The "unification of current theoretical properties" is overstated. The two referenced papers (Zimmermann et al.; Kügelgen et al.) are quite similar in that they show identifiability results for multi-view nonlinear ICA and related settings. Specifically, Kügelgen et al. already discuss the connection of their work to the work from Zimmermann et al.*
>
> We believe the reviewer refers to the two sentences in von Kügelgen et al. (2021): "Moreover, (1) can be interpreted as alignment (numerator) and uniformity (denominator) terms, the latter constituting a nonparametric entropy estimator of the representation as $K \to \infty$ [124]." and "As discussed in § 2, contrastive SSL with negative samples using InfoNCE (1) as an objective can asymptotically be understood as alignment with entropy regularisation [124], i.e., objective (5).", where they relate to Wang and Isola (2020) and therefore by extension to Zimmermann et al. (2021).
>
> If this is what the reviewer is referring to, we believe that this is not a direct relationship. The practical implementation of the uniformity term in Zimmermann et al. (2021) is equivalent to a KDE estimation of the entropy, and tends to the entropy at the rate stated in our Theorem 2. In Von Kügelgen et al. (2021), they consider the real entropy for their theorem. So the connection in (Von Kügelgen et al. 2021) with (Zimmermann et al. 2021) only holds in the asymptotic regime.
>
> Conversely, we prove that both results hold when maximising the lower bound (1) on the mutual
> information, without using the uniformity loss but the entropy directly. That is, both results are
> unified through mutual information maximisation. Therefore, the connection in our analysis holds
> without the need of employing neither the uniformity loss nor a KDE estimation of the entropy.
>
> We have clarified this in the revised version, dedicating Appendix A3 to it. Thank you for letting us know about this potential source of confusion.

---

> > ### Comment · Reviewer_KfpZ · 2022-11-22
> > **Reply to the authors**
> >
> > I appreciate the response of the authors.
> >
> > The authors did not address my concern that the "improved robustness of one of the proposed models is not sufficiently clear from the experiments" and I agree with reviewer r5LV in that the empirical evaluation is still insufficient.
> >
> > I still believe that the decomposition of the MI and estimation of the two individual terms (cross-entropy and entropy) is an interesting research direction, however, I think in its current form the paper is not yet ready for publication and requires a major revision to make the contribution clearer and the empirical evaluation more thorough. The authors might even consider separating the unification of different theoretical properties and the estimation of individual entropy terms into two separate papers to make the contributions clearer and more self-contained.
> >
> > Regarding previous work on the recovery of the true latent variables, I would like to point out further relevant work related to multi-view nonlinear ICA and self-supervised learning that could be put into context:
> >
> > - Luigi Gresele, Paul K. Rubenstein, Arash Mehrjou, Francesco Locatello, Bernhard Schölkopf: The Incomplete Rosetta Stone problem: Identifiability results for Multi-view Nonlinear ICA. UAI 2019: 217-227
> > - Nonlinear Multiview Analysis: Identifiability and Neural Network-Assisted Implementation. IEEE Trans. Signal Process. 68: 2697-2712 (2020)
> > - Qi Lyu, Xiao Fu, Weiran Wang, Songtao Lu: Understanding Latent Correlation-Based Multiview Learning and Self-Supervision: An Identifiability Perspective. ICLR 2022

---

### Official Review · Reviewer_r5LV · 2022-10-24

**Confidence:** 4
**Correctness:** 1
**Technical Novelty And Significance:** 2
**Empirical Novelty And Significance:** 1
**Recommendation:** 1

**Clarity, Quality, Novelty And Reproducibility:**

Overall the paper is reasonably well written and flows quite well. However, it suffers a lot from juggling between precise and verifiable st
atements and loose/handwavy explanations. For instance:
- 'we show that many of these approaches are maximising an approximate lower bound...': while this may hold true in the case of contrastive methods, the content of the paper definitely does not **show** in any reasonable mathematical sense that BYOL or DINO maximises mutual information.
- 'In this section we demonstrate ...': Same here, it is still unclear after reading this section how BYOL and DINO relate to mutual informa
tion maximization.


**Strength And Weaknesses:**

**Strength:**
- The paper tackles a potentially interesting (though quite thoroughly studied) topic.
- The quality of the writing is above average.

**Weaknesses:**

__Theory:__
- The theoretical contributions of the paper consists in showing that different self-supervised learning approaches all aim at maximizing the mutual information. This contribution falls short for several reasons:
  - Contrastive approaches are already known to approximate a lower bound on the mutual information even though this lower bound mostly holds when both views come from the same distribution. The paper provides slightly more precise, but rather straightforward analysis for when views come from different distributions.
  - The analysis for both BYOL and DINO is extremely handwavy: there is no clear explanation for why and how either of these algorithms would maximise the entropy term. In their current form, these paragraphs provide no precise statement relating BYOL's or DINO's objective to mutual information maximization, and I would argue in favor of removing them.
- I fail to understand what makes EntRecCont significantly different from CMC/SimCLR; in the case where $P_{Z_1} = P_{Z_2}$. When we add back the missing - in the definition of the entropy and remove the constant terms inside of the logarithm, we fall back to a standard NTXEnt loss (potentially up to normalization of the projections). Given how close the two are, it seems necessary to at least mention the equivalence/similarity.
- I fail to understand the usefulness of mentionning Theorem 1, given both that the KDE approximation has clear limitations, probably making
 it a very bad approximation of Equation (1) (typically the provided lower bound cannot approximate Mutual Information higher than O(ln(k)), where k is the number of 'negatives' used, as mentionned in (McAlester et al. 2018)), and there are no experimental validation that any of the methods mentionned either verifies the hypothesis of the theorem, or matches their implications.

__Experiments:__

The experimental part requires a significant rework, as it currently gives a biased picture of the performance of the algorithms. More precisely:
  - The performance of BYOL in Table 1 is **significantly** lower than the performance reported in (Grill et al. 2020), namely the performance at 300 epochs is claimed to be 65.5% top-1, while the original paper mentions 72.5% **using the same architecture**. This changes significantly the story told by Table 1. While the difference may be explained by a change of hyperparameters, or image transformations, it seems disingenuous to place the algorithm in a setting that does not allow it to provide its best performance. I would suggest both putting back the original results in the table, and revising the claim of near SOTA performance.
  - Similarly, the results presented in Table 2 strongly contradicts results presented in (Grill et al. 2020), where BYOL loses less than 1% top-1 accuracy when going from batch size 4096 to 512, as opposed to 34.2%(??) in this paper. Again this significantly alters the conclusion provided by the paper.

**Summary Of The Paper:**

This paper proposes to unveil some relationships between mutual information maximization and multi-view self-supervised learning.
It claims to show that many SSL approaches can be framed as maximizing mutual information, and provide yet another approach that
considers maximizing mutual information as its objective (in an arguably 'more general' way). It then provides some empirical
results using two variants of this method.


**Summary Of The Review:**

The paper at hand provides a new take on the relationship between multi-view self-supervised learning and mutual information maximization. While the topic is interesting, the paper falls short in many directions. The main theoretical contributions of the paper are either already known (contrastive methods optimize a lower bound on the mutual information) or hardly made precise enough to be useful (non-contrastive methods e.g. BYOL and DINO optimize a lower bound on the mutual information).

I am more than doubtful of the empirical results: the results of at least one of the baselines provided are drastically below the results provided in the corresponding paper (BYOL @300 epochs is mentionned to achieve 65.8%, the original paper mentions 72.5%), and the batch size analysis displays the same trend (the paper claims that BYOL loses 34.2% accuracy when trained with batch size 512, while the original paper mentions that the loss in accuracy is about 1%).

Given both points, I strongly recommend rejecting this paper.

---

> ### Author Response · Authors · 2022-11-17
> **Answer to Reviewer r5LV [1/3]**
>
> We would like to thank the reviewer for their review, it gave us the opportunity to clarify certain aspects that will aid the readability for future readers. Please, find our answers to your comments below.
>
> * *Contrastive approaches are already known to approximate a lower bound on the mutual information even though this lower bound mostly holds when both views come from the same distribution. The paper provides slightly more precise, but rather straightforward analysis for when views come from different distributions.*
>
> We appreciate the comment and the opportunity to clarify our work. To the best of our knowledge,
> it was only known that InfoNCE-like methods to contrastive learning were maximising a lower bound to the mutual information. A crucial part of these existing analyses for these methods is that they consider that the negative samples are iid. However, this excluded SimCLR, since for a sample $Z_1^{(i)}$ the negative samples $Z_1^{(j)}$ and $Z_2^{(j)}$ are neither independent nor identically distributed.
>
> Hence, even if a connection between SimCLR and mutual information maximisation could be intuitive for some, this formalisation and the lifting of both the independence and identically distributed assumptions was not present in the literature. Thus, we believe that having it explicitly written is a contribution to the community.  If we are missing additional references please do let us know and we would be glad to review them.
>
> We are glad the presentation was clear. It was our intention to disseminate the understanding of these phenomena in the simplest possible manner.
>
> * *The analysis for both BYOL and DINO is extremely handwavy: there is no clear explanation for why and how either of these algorithms would maximise the entropy term. In their current form, these paragraphs provide no precise statement relating BYOL's or DINO's objective to mutual information maximization, and I would argue in favor of removing them.*
>
> Thank you for the suggestion, please allow us to clarify why we believe these results are valuable. One of our contributions is to show that the contrastive family, plus Deep Cluster and SwAV, all maximise mutual information (section 2 and appendix d) while for other algorithms (among which BYOL and DINO) one cannot always make the same claim. Moreover, we show that empirically this difference has an effect on the robustness of the algorithms, namely, those that maximise mutual information empirically seem more robust to individual changes of hyper parameters.
>
> Note how we do not say that these algorithms maximise the entropy term. The precise statement we make is "To avoid collapse in the absence of negative pairs, they have to employ different engineering techniques that as we show can help to maintain a high entropy term in (1) in different ways". These methods aim at **maintaining** a high entropy, in order to prevent collapse (if entropy was not at least maintained then collapse could happen), not maximising it.
>
> To understand more about these algorithms we showed how the moving average parameter $\lambda$ navigates between complete independence (and hence having a **fixed**, not maximised, entropy that maximises the mutual information) and complete dependence (and hence leading to collapse). This gives an interpretation of this parameter and why having a large $\lambda$ so that the parameters are close to independence is crucial, which aligns with the experiments in Grill et al. (2020) and Tables 1 and 2.
>
> Regarding DINO, we also explain that in addition to the moving average parameter they also have a parameter for the centring $\mu$. The smaller $\mu$ is, the higher the conditional entropy $H(W_2|Z_2)$, and therefore by the fact that conditioning reduces the entropy, the higher $H(W_2)$, as desired to maximise $I(Z_1;W_2) \leq I(Z_1;Z_2)$. This gives DINO more stability than BYOL. However, if this centring is totally removed with $\mu \to 1$ then the model collapses since high $H(W_2|Z_2) \leq H(W_2)$ is not enforced. This is also in line with Caron et al. (2021) results (their Appendix D).
>
> These analyses create a connection between mutual information maximisation (through maintainance of the entropy, not maximisation) and these two methods. We agree with the reviewer that it is not totally formalised and we made sure to highlight this in the revised version. What exactly these algorithms do is still to be fully understood, but our analysis is an attempt to identify patterns and generate hypotheses that could push the community to unveil some of the remaining questions.

---

> ### Author Response · Authors · 2022-11-17
> **Answer to Reviewer r5LV [2/3]**
>
> * *I fail to understand what makes EntRecCont significantly different from CMC/SimCLR; in the case where $P_{Z_1} = P_{Z_2}$. When we add back the missing - in the definition of the entropy and remove the constant terms inside of the logarithm, we fall back to a standard NTXEnt loss (potentially up to normalization of the projections). Given how close the two are, it seems necessary to at least mention the equivalence/similarity.*
>
> Thank you for bringing up this potential source of confusion. We agree that in the case where $P_{Z_1} =P_{Z_2}$ EntRecCont would be equivalent to CMC (up to constants), we will follow the suggestion and make sure to explicitly mention this equivalence.  On the other hand, the equivalence does not hold true for SimCLR since the negative samples are not independent of each other.
>
> That said, the objective of EntRec is not to claim to be completely different from CMC (or SimCLR) nor to beat state-of-the-art benchmarks, but to show that a very direct and simple way to maximise mutual information leads to competitive results.
>
> It is worth noting that $P_{Z_1}$ and $P_{Z_2}$ are not equal when calculating (2), (3), or (4). The reason is that at each iteration $Z_1 = f \circ t_1 (X)$ and $Z_2 = f \circ t_2 (X)$ where $t_1$ and $t_2$ are sampled such that they are different, which is the point of the multi-view setting. It is not until training starts to converge and the encoding of different views results in projections such that ${Z_1} \approx {Z_2}$. If these distributions were close since the beginning of training, it would mean that the encoder and the projection head are already invariant to the view transformations, which is one of the appeals of multi-view SSL.
>
> * *I fail to understand the usefulness of mentionning Theorem 1, given both that the KDE approximation has clear limitations, probably making it a very bad approximation of Equation (1) (typically the provided lower bound cannot approximate Mutual Information higher than O(ln(k)), where k is the number of 'negatives' used, as mentionned in (McAlester et al. 2018)), and there are no experimental validation that any of the methods mentionned either verifies the hypothesis of the theorem, or matches their implications.*
>
> Thank you for the comment, please let us shine more light on Theorem 1. Theorem 1 is a generalisation of the objective presented in Zimmermann et al. (2021) and can be employed to recover their main results. Theorem 1 also includes the results from Von Kügelgen et al. (2021). Specifically, it shows that these results arise when maximising the mutual information lower bound from (1). We believe that both these works are useful, and for this reason, we think this unification under a maximisation of mutual information is also helpful. Note also that Theorem 1 does not require that the entropy is estimated using a KDE.
>
> As mentioned in the paper, when the KDE estimation and the reconstruction are done with densities of the form $e^{- \rho(z_1,z_2)}$ then EntRecCont recovers the alignment and uniformity loss in Zimmermann et al. (2021). Similarly, if $\rho(z_1,z_2) = \lVert z_2 - z_1 \rVert^2$ then it recovers the loss employed in Von Kügelgen et al. (2021). We agree with the reviewer that empirical support would further strengthen the overall publication and we will consider this in future works.
>
> * *The performance of BYOL in Table 1 is **significantly** lower than the performance reported in (Grill et al. 2020), namely the performance at 300 epochs is claimed to be 65.5% top-1, while the original paper mentions 72.5% **using the same architecture**. This changes significantly the story told by Table 1. While the difference may be explained by a change of hyperparameters, or image transformations, it seems disingenuous to place the algorithm in a setting that does not allow it to provide its best performance. I would suggest both putting back the original results in the table, and revising the claim of near SOTA performance*
>
> The reviewer is correct, the results we obtained were inferior to those previously published.
>
> On the one hand, we are happy to report that after further hyper-parameter tuning we were able to reach, and even surpass, the results reported by (Grill et al. 2020): we have now achieved 73.8% accuracy in 400 epochs. The differences indeed begin in hyper parameters settings. We have updated the paper and explicitly called out that these results are the ones we reproduced, so readers can distinguish them from those of the original publication.
>
> On the other hand, what is important for our analysis is not the absolute accuracy but the speed of degradation when a single hyper parameter is changed. Even with the new results, the conclusion remains unchanged: BYOL (and DINO) degrade their performance the most when the batch size is changed and all other parameters are left unchanged. In the case of BYOL the accuracies are the following (see next comment).

---

> ### Author Response · Authors · 2022-11-17
> **Answer to Reviewer r5LV [3/3]**
>
> | batch size     | BYOL Top 1 Accuracy |
> | ----------- | ----------- |
> | 4096      | 73.8       |
> | 2048   | 71.7        |
> | 1024      | 51.8       |
> | 512   | 31.3        |
>
> | # of epochs      | BYOL Top 1 Accuracy |
> | ----------- | ----------- |
> | 400      | 73.8       |
> | 300   | 72.0        |
> | 200      | 68.8       |
> | 100   | 60.4        |
>
> * *Similarly, the results presented in Table 2 strongly contradicts results presented in (Grill et al. 2020), where BYOL loses less than 1% top-1 accuracy when going from batch size 4096 to 512, as opposed to 34.2%(??) in this paper. Again this significantly alters the conclusion provided by the paper.*
>
> Thank you for highlighting this, as it is indeed one of the main conclusions of our analysis and it is important to explain the difference between the results of (Grill et al. 2020) and ours.
>
> Grill et al. (2020) reported the results when changing the batch size and as stated in the original work (section 5)  “To avoid re-tuning other hyperparameters, we average gradients over $N$ consecutive steps before updating the online network when reducing the batch size by a factor $N$. The target network is updated once every $N$ steps, after the update of the online network; we accumulate the $N$ steps in parallel in our runs.". This is a legitimate procedure but it goes beyond the point of our analysis. With all the algorithms we compared we acted by only changing the batch size and leaving all the other parameters and training protocol unchanged. This is why our results are very different from (Grill et al. 2020). We are not interested in the maximum performance one can obtain, but in the robustness of the algorithm to single hyper-parameters changes. Note that we are not implying that BYOL (or DINO, or any other algorithm) is worse than others. We are simply showing a common pattern that emerges in SSL frameworks when the entropy is not explicitly maximised. Thank you for your comment that gave us the opportunity to clarify this important point.
>
> * *Overall the paper is reasonably well written and flows quite well. However, it suffers a lot from juggling between precise and verifiable st atements and loose/handwavy explanations. For instance:*
>   * *'we show that many of these approaches are maximising an approximate lower bound...': while this may hold true in the case of contrastive methods, the content of the paper definitely does not show in any reasonable mathematical sense that BYOL or DINO maximises mutual information.*
>   * *'In this section we demonstrate ...': Same here, it is still unclear after reading this section how BYOL and DINO relate to mutual informa tion maximization.*
>
> Thank you for pointing out that this might not be clear.
>
> As the reviewer says, this holds true for the contrastive methods as well as for DeepCluster and SwAV.
>
> To better understand BYOL and DINO we showed that the moving average parameter $\lambda$ navigates between complete independence (and hence having a fixed entropy) and complete dependence (and hence leading to collapse). This gives an interpretation of this parameter and why having a large $\lambda$ so that the parameters are close to independence is crucial, which aligns with the experiments in Grill et al. (2020) and Tables 1 and 2.
>
> We also note how in DINO, in addition to the moving average parameter, there is a  centring parameter $\mu$. The smaller $\mu$ the higher the conditional entropy $H(W_2|Z_2)$, and therefore, by the fact that conditioning reduces the entropy, the higher $H(W_2)$, as desired to maximise $I(Z_1;W_2) \leq I(Z_1;Z_2)$. This gives DINO more stability than BYOL as shown in our experiments. However, if this centring is removed with $\mu \to 1$ then the model collapses since high $H(W_2|Z_2) \leq H(W_2)$ is not enforced. This is also in line with Caron et al. (2021) results (their Appendix D).
>
> These analyses create a connection between mutual information maximisation and these two methods. We agree with the reviewer that it is not completely formalised and we made sure to highlight this in the revised version. What exactly these algorithms do is still to be fully understood, but our analysis is an attempt to identify patterns and generate hypotheses that could push the community to unveil some of the remaining questions.

---

> > ### Comment · Reviewer_r5LV · 2022-11-18
> > **Answer to rebuttal**
> >
> > After reading the rebuttal, I think that most of my concerns about the paper remain:
> > - I still think the contribution on showing that SimCLR and related contrastive methods do not __exactly__ optimize the mutual information lower bound is quite weak.
> > - I still think the contribution on other algorithms (BYOL, DINO, Deep Cluster, ...) lacks a proper formal grounding, and I don't think the intuitions provided bring that much insight.
> > - I still think the experimental analysis is lacunary, and might even mislead readers. To extend on that point,  my main concerns on the experiments was that one of the baseline seemed to have been severely under-tuned, which greatly altered the conclusions of the paper. While the rebuttal seems to indicate that after tuning this baseline better, the overall performance comparison is altered, the authors still mention that the results on the batch size remain an interesting finding. I disagree with this statement. BYOL is to be considered robust to changes in batch size, in the sense that, if you follow the procedure presented in the paper, and you modify the batch size, the algorithm should still perform well. This is, to me, the only practically useful definition of robustness, as it directly provides practitioners with a way to make the algorithm work for any batch size **at no additional computational cost** and at **very little additional implementation cost**. This is different from an algorithm that would require additional hyperparameter tuning to perform well. Comparing 'everything else remaining fixed' does not provide, to me, any useful notion of robustness, given that, if I provided another basis for the hyperparameters used (say for instance parameterizing the learning rate as eta * batch_size, and used eta and batch_size as my two hyperparameters), the results would potentially be extremely different. In a nutshell, to have a fair comparison in term of robustness with BYOL, one should follow the procedure provided in (Grill et al. 2020) that allows for scaling the batch size. Since this procedure is not followed, I still don't consider the robustness results to be of practical interest.

---

### Official Review · Reviewer_Nfqh · 2022-10-28

**Confidence:** 4
**Correctness:** 3
**Technical Novelty And Significance:** 3
**Empirical Novelty And Significance:** 2
**Recommendation:** 5

**Clarity, Quality, Novelty And Reproducibility:**

The manuscript is easy to follow. It seems not straightforward to implement EntRec.

**Strength And Weaknesses:**

Strength:
1. The manuscript provides a good summary to previous multi-view SSL methods.
2. It is interesting to see that a straightforward approximation of entropy (with traditional kernel density estimator or discrete entropy) and reconstruction can achieve comparable performances.

Weaknesses:
1. The main result that multi-view (augmentation-based) SSL methods maximize MI has already been published in previous work, e.g.,
https://arxiv.org/abs/2010.02037. For acceptance, I would expect the authors to provide convincing arguments on why their work improve over previous work in this regard.
2. A large part of the paper reads as a report covering previous approaches to SSL, making it difficult to see the actual contribution of this work over previous work.
3. There are very few experiments included in the paper. Moreover, current result is very weak and does not demonstrate the advantage of EntRec:
3.1. The results show that the proposed methods perform worse than DINO (ICCV’21) which is already a year old. If the proposed methods are principled and direct approaches to the same objective which all others minimize, why do they perform worse?
3.2. The authors claim that the low bound in Eq. (1) helps obtain good representation and separate semantics from irrelevant information.
However, it is not verified in detail.

**Summary Of The Paper:**

This paper proposes a general framework to understand multi-view self-supervised learning (SSL) methods from a mutual information maximization perspective. Authors then use a naive lower bound of MI, i.e., entropy + reconstruction, as the objective function. The proposed methods, namely EntRecCont and EntRecDisc, achieve comparable performances to baselines.

**Summary Of The Review:**

This paper contributes some interesting ideas and also provides a good summary to previous literature. However, the theoretical contribution seems marginal. The experiment is also a bit weak.

---

> ### Author Response · Authors · 2022-11-17
> **Answer to Reviewer Nfqh [1/2]**
>
> We would like to thank the reviewer for taking the time to review our paper and provides us with valuable feedback. Please, find below our answers to your comments.
>
> * *The main result that multi-view (augmentation-based) SSL methods maximize MI has already been published in previous work, e.g., https://arxiv.org/abs/2010.02037. For acceptance, I would expect the authors to provide convincing arguments on why their work improve over previous work in this regard.*
>
> Thank you for allowing us to clarify this aspect. The main difference with respect to already published works is that we extend the analysis to include algorithms such as SimCLR, DeepCluster, SwAV, DINO, or BYOL for which the previous analysis did not hold.
>
> We agree that in the literature it is known that the InfoNCE loss is a lower bound on the mutual information as shown by Poole et al. (2019). The result from Wu et al. (2020) suggested by this reviewer says that the samples from the denominator of the InfoNCE, as long as they are iid, can come from a different distribution. We were not aware of this article and we added it to the revised version although in practice (Wu et al 2020) suffer from the same limitations of (Poole et al. 2019). Specifically, they both assume iid of the negative samples, assumptions that are violated in methods such as SimCLR. In SimCLR the negative samples are not iid since $Z_1^{(j)}$ and $Z_2^{(j)}$ are neither independent nor identically distributed. In particular, we see how they relate to a maximisation of the mutual information with a penalty of the form $D_{\textnormal{JS}} (P_{Z_1} \lVert P_{Z_2})$, where $D_{\textnormal{JS}}$ is the Jensen-Shannon divergence. As far as we know, this clarification and connection between mutual information and SimCLR are not known in the literature.
>
> Additionally, we include an analysis of how DeepCluster and SwAV also maximise the mutual information of the projections through a surrogate discrete random variable, similarly to DINO; and give an argument also tying BYOL and SimSiam with this maximisation. As far as we know, none of these connections exist in the literature.
>
> * *A large part of the paper reads as a report covering previous approaches to SSL, making it difficult to see the actual contribution of this work over previous work.*
>
> Thank you for letting us know about this. We have now clarified more explicitly the contributions of our work, in summary:
>
> * We provide an analysis of a few previously published works in SSL tying them together through the lens of information theory. In particular, they aim to maximise a lower bound on the mutual information. While this was known for InfoNCE-based algorithms, our analysis is applicable to more algorithms.
> * We show that maximising a certain lower bound on the mutual information has the properties of being able to recover the true latent variables and to separate content from style, unifying the theory from (Zimmermann et al. 2021) and (Von Kügelgen et al. 2021).
> * We observed that none of the existing algorithms maximises this mutual information lower bound in a direct way, so we proposed such an algorithm and evaluated its performance.
> * We empirically show that different families of algorithms (contrastive vs reconstruction-based) exhibit different levels of robustness with respect to hyper-parameter choices. Namely, those that attempt to maximise the entropy directly show better robustness with respect to changes of single hyper-parameters such as batch size and training epochs.
> * We showed that the proposed algorithm performs on par with existing contrastive algorithms while being simpler than previous works.
> * We acknowledge that the InfoNCE connection to the mutual information and how it was proved by Poole et al. (2019) is not mentioned until Section 3.1. We did not notice we missed that, we clarified this in the revised version.

---

> ### Author Response · Authors · 2022-11-17
> **Answer to Reviewer Nfqh [2/2]**
>
> * *There are very few experiments included in the paper. Moreover, current result is very weak and does not demonstrate the advantage of EntRec: 3.1. The results show that the proposed methods perform worse than DINO (ICCV’21) which is already a year old. If the proposed methods are principled and direct approaches to the same objective which all others minimize, why do they perform worse? 3.2. The authors claim that the low bound in Eq. (1) helps obtain good representation and separate semantics from irrelevant information. However, it is not verified in detail.*
>
> It is true that the proposed algorithm does not perform better than DINO in terms of best accuracy. We note, however, that one of the contributions is to show how some methods (including DINO) are more fragile than others with respect to hyper parameters turning. For example, when only the batch size was changed the performance of DINO degraded the most, from 71.6 to 4.6, while other algorithms such as EntRec degraded at a much lower rate, from 69.7 to 65.5 (Cont) or 66.9 to 63.0 (Disc).
>
> Additionally, while DINO has a 60000-sized head size, and uses full precision for training to boost performance, EntRec uses 128-sized head size and half-precision for training.
>
> Another objective of the paper was to showcase how previous approaches are aiming to maximise the mutual information and how a naive, more direct approach (in this case the implementations of EntRecCont and EntRecDisc) could be competitive with these current methods. But it was not the intention to provide a new benchmark-beating algorithm.
>
> Other contributions are theoretical and are listed above and in the revised paper. We hope this reviewer shares our belief that not all research has to necessarily lead to bold numbers in order to advance the machine learning field. We believe our answers clarified what was unclear, if this is not the case we hope more detail about what else needs to be clarified can be shared with us by this reviewer.
>
> In any case, we are grateful for the review and are confident the clarifications requested have made our paper stronger.

---

### Official Review · Reviewer_D75g · 2022-10-30

**Confidence:** 3
**Correctness:** 3
**Technical Novelty And Significance:** 2
**Empirical Novelty And Significance:** 2
**Recommendation:** 5

**Clarity, Quality, Novelty And Reproducibility:**

Overall, the manuscript is well-organized, but some parts can be made clearer (see weakness). The contributions are not significant since similar loss functions have been studied and the main theoretical results are not new. In terms of reproducibility, the detailed algorithm could be given for better clarity.

**Strength And Weaknesses:**

**Strength**:

It is an interesting idea to generalize previous model identification results of different generative models by using the mutual information lower bound. The authors also provide detailed analysis showing that many existing multiview SSL algorithms approximately maximize such a lower bound, which could help deepen the understanding of different SSL paradigms. However, the theoretical aspects as well as the empirical ones are not sufficiently clear in its current version.

**Weaknesses**:
1. The novelty seems not significant since a similar loss has been proposed and analyzed in Wang and Isola (2020) as uniformity and alignment terms. The uniformity is the entropy term here while the alignment corresponds to the reconstruction term (or cross entropy term). Could the authors spell out the differences and the advantages?
2. It is unclear that how Eq. (1) unifies both model identification theorems of Zimmermann et al. (2021) and Von Kugelgen et al. (2021). Specifically, contrastive loss (i.e., InfoNCE loss) is analyzed in Zimmermann’s work while $l_2$ matching loss is used in Von Kugelgen’s work. How is the equivalence between such a loss and (1) established? It is hard to tell how the proof techniques here differ from those of previous works.
3. The convergence rate results of the proposed methods are interesting but not very clear. Could the authors give more details on how the results are derived and perhaps more explanation?
4. It would help readers better understand if the detailed algorithm is listed.
5. It unclear how useful the proposed method is in practice since kernel density estimation is required for each iteration. Is there a wall time or complexity comparison?
6. In terms of experiment results, it might be more straightforward using synthetic experiments to showcase the desired theoretical properties. For example, how well the mutual information is maximized? How does that help identify the true latent variables and/or separate the content from the style, compared to the baselines? Is it necessary to learn good representations only if a tight bound is maximized?

Some minor comments and questions:
1. Both Zimmermann et al. (2021) and Von Kugelgen et al. (2021) assume that the generative function is the same for each view. As a comparison, in [1], it is shown that the generative function $g(\cdot)$'s do not need to be identical (in a more natural multivew setting) for each view in order to recover the true latent variables. Does the proposed criterion lead to similar model identification results under such a setting?
2. The interpretation in the first paragraph on page 3 is not very clear to the reviewer. Could the authors be more specific about what information can be and is learned is? It seems the entropy term is simply a constrain (could be other alternatives) to avoid degenerated solutions.
3. Why is the proposed method more robust to changes of hyper-parameters, compared to other baselines?
4. How is the parameter of kernel band width selected? Or how the density is estimated?
5. In the Conclusion section, typo ‘or or isolating’.

[1] Qi Lyu, Xiao Fu, Weiran Wang, and Songtao Lu. "Understanding latent correlation-based multiview learning and self-supervision: An identifiability perspective." In International Conference on Learning Representations. 2021.


**Summary Of The Paper:**

This manuscript provides an information theoretical perspective for multiview self-supervised learning (SSL). In specific, the authors show that a lower bound of mutual information is a useful objective function to learn informative representations. Additionally, optimizing such an objective function could be regarded as a generalization of both learning true latent variables (as in Zimmermann et al. (2021)) and separating content information from the irrelevant (as in Von Kugelgen et al. (2021)), from a generative model perspective.

Besides, the work also analyzes the loss functions of different SSL methods, e.g., contrastive learning based and projection reconstruction based ones, and demonstrates that these approaches do not maximize the mutual information lower bound exactly.


**Summary Of The Review:**

This paper gives an interesting point of view for analyzing SSL using mutual information lower bound and proposes a method based on the criterion. However, both the theoretical and empirical results are not sufficiently clear.

---

> ### Author Response · Authors · 2022-11-17
> **Answer to Reviewer D75g [1/3]**
>
> We would like to thank the reviewer for their detailed review and constructive feedback. Please, find below the answers to your comments.
>
> * *The novelty seems not significant since a similar loss has been proposed and analyzed in Wang and Isola (2020) as uniformity and alignment terms. The uniformity is the entropy term here while the alignment corresponds to the reconstruction term (or cross entropy term). Could the authors spell out the differences and the advantages?*
>
> Thank you for giving us the opportunity to clarify the existing differences between our work and previously published articles. Specifically, the main differences between our work and that of Wang and Isola (2020) are that (i) we prove that Wang and Isola (2020)’s loss is a particular case of (1);  (ii) we formalize the relationship between their loss (in a more general sense) with the mutual information; and (iii) we show that (1) also includes other SSL methods, which is not the case for Wang and Isola (2020). We expand on these differences and improvements in the rest of the answer.
>
> In particular, with a KDE approximation of the entropy (EntRecCont) with a Gaussian kernel and a reconstruction density $q_{Z_2|Z_1=z_1}(z_2)\propto e^{-t \lVert Z_1 - Z_2 \rVert^\alpha}$ this recovers the alignment-uniformity loss from Wang and Isola (2020). The same happens with the extension of the alignment-uniformity loss in Zimmermann et al. (2021) with a  KDE approximation of the entropy with a kernel and a reconstruction density of the form $q_{Z_2|Z_1=z_1}(z_2)\propto e^{- \rho(z_1,z_2)/\tau}$.
>
> Then the relationship with mutual information is formalised compared to Wang and Isola (2020). First, it is acknowledged that the density estimation is Joe's (1989) and not Ahmad and Lin's (1976), since the random variables lie in $\mathbb{R}^d$. Second, it is noted that when considering the cross entropy, the EntRecCont loss tends to a proper bound on the mutual information at the rate specified in the paper. In contrast, in Wang and Isola (2020), the estimation of the entropy is said to be also an estimation of the mutual information since the encoder is deterministic, which is not true for the differential entropy in the case of continuous random variables.
>
> A benefit of the entropy and reconstruction interpretation of the mutual information bound is that other contrastive methods which do not use the InfoNCE loss, e.g. SimCLR, can be included under this framework. Wang and Isola (2020) and Zimmermann et al. (2021) only consider methods where the contrastive loss is of the form of their equation (1). This excludes important contrastive learning methods such as SimCLR, since in the denominator for projection $Z_1^{(i)}$ one would have as negative examples both $Z_1^{(j)}$ and $Z_2^{(j)}$ for $j \neq i$, which clearly are not i.i.d..
>
> Furthermore, this interpretation also includes methods that employ a surrogate discrete variable after the projection such as DeepCluser, SwAV, or DINO.
>
> * *It is unclear that how Eq. (1) unifies both model identification theorems of Zimmermann et al. (2021) and Von Kügelgen et al. (2021). Specifically, contrastive loss (i.e., InfoNCE loss) is analyzed in Zimmermann’s work while matching $l_2$ loss is used in Von Kügelgen’s work. How is the equivalence between such a loss and (1) established? It is hard to tell how the proof techniques here differ from those of previous works.*
>
> This is a great question, we answer this question here and provide a similar answer in the revised version of the paper as it is an important point.
>
> In (Zimmermann et al., 2021), for the identification theorems to work, the reconstruction density needs to be of the form $q_{Z_2|Z_1=z_1}(z_2) = C_h(z_1) e^{- \alpha \rho(z_1,z_2)}$ for some metric $\rho$. Showing that maximising (1) maintains the properties of (Zimmermann et al., 2021) requires noting that their Proposition 4 states that if only the cross entropy is maximised then $\rho(\tilde{z}_1, \tilde{z}_2) = \alpha \rho(h(\tilde{z}_1), h(\tilde{z}_2))$ and $C(\tilde{z}_1) = C_h(\tilde{z}_1)$. Then, one sees that if this happens then $H(Z_2) = H(\tilde{Z}_2) = \log |\mathcal{Z} |$ (where $Z_2$ is the projection and $\tilde{Z}_2$ the real latent variable) and hence the entropy is also maximised. Therefore, the unique family of maximisers of the reconstruction term that recover the latent variables up to affine transformations are maximisers of the entropy, and hence are the unique family of maximisers of the mutual information.
>
> For Von Kügelgen's et al. (2021) theorem to work the reconstruction density needs to be further restricted to be of the form $q_{Z_2|Z_1=z_1}(z_2) = C_h(z_1) e^{- \alpha \lVert z_1 - z_2 \rVert^2}$, i.e., $\rho(z_1,z_2) = \lVert z_1 - z_2 \rVert^2$. Showing that maximising (1) maintains the properties of Von Kügelgen's et al. (2021) only requires noting that the reconstruction term reduces to the $\ell_2$ loss with a Gaussian reconstruction density.

---

> ### Author Response · Authors · 2022-11-17
> **Answer to Reviewer D75g [2/3]**
>
> * *The convergence rate results of the proposed methods are interesting but not very clear. Could the authors give more details on how the results are derived and perhaps more explanation?*
>
> Thank you for allowing us to clarify this aspect. We now included a section in the Appendix (the new Appendix F) detailing the bias and variance terms that determine the convergence in MSE of the estimators.
>
> * *It would help readers better understand if the detailed algorithm is listed.*
>
> Thank you for the suggestion. Indeed we agree that an algorithm would help. We included this in the revised version in the new Appendix E.
>
> * *It unclear how useful the proposed method is in practice since kernel density estimation is required for each iteration. Is there a wall time or complexity comparison?*
>
> Thank you, we agree that this could be a concern and we provided the information reported here to show that in practice EntRec is on-par with other algorithms in terms of complexity.
>
> Specifically, the complexity of the EntRecCont using KDE is the same as for the other contrastive methods like CMC and SimCLR, i.e. $\mathcal{O}(k^2 d)$. This can be seen by observing the CMC and SimCLR losses in the forms of (10) and (13) and comparing it with the EntRecCont loss in (5).
>
> On the other hand, the complexity of the discrete version is only $\mathcal{O}(kd)$.
>
> Thank you for this comment, we agree that the additional information provided in the revised version further solidifies the paper.
>
> * *Both Zimmermann et al. (2021) and Von Kügelgen et al. (2021) assume that the generative function is the same for each view. As a comparison, in [1], it is shown that the generative function's do not need to be identical (in a more natural multivew setting) for each view in order to recover the true latent variables. Does the proposed criterion lead to similar model identification results under such a setting?*
>
> Unfortunately, not immediately. An important part to show that maximising (1) maintains the properties of Zimmermann et al. (2021) is to use their Proposition 4. This requires that $h = f \circ g$ is an isometry, that is $\rho(z_1, z_2) = \rho(h(z_1), h(z_2))$. In the case that we have two generative processes, $g_1$ and $g_2$, leading to two different $h_1$ and $h_2$ the properties from isometries used there would not be applicable.
>
> The proposed setting is interesting and could be possible, but new theories should be developed for that. Thank you for the suggestion, this can be an exciting topic for further investigations.
>
> * *The interpretation in the first paragraph on page 3 is not very clear to the reviewer. Could the authors be more specific about what information can be and is learned is? It seems the entropy term is simply a constrain (could be other alternatives) to avoid degenerated solutions.*
>
> Great point. Maximising the entropy indeed helps prevent degenerate solutions. Maximising the entropy, however, is more than a constraint but a measure of the "available" information.
>
> The entropy is a measure of the uncertainty of a random variable. The mutual information $I(Z_1;Z_2)$, defined as $H(Z_2) - H(Z_2 | Z_1)$ is then a measure of the reduction in uncertainty on the random variable $Z_2$ after observing $Z_1$. For this reason, we can informally interpret the entropy term as a measure of how much information can be learned (the higher the better) and the cross entropy (or the reconstruction) as how much information is actually learned.
>
> Note that this interpretation, for continuous random variables, is only possible when the two are compared together since differential entropy allows negative values and only their difference constitutes "information".
>
> * *Why is the proposed method more robust to changes of hyper-parameters, compared to other baselines?*
>
> This is a great question. The exact reason cannot be stated with complete certainty. From comparing the robustness of different algorithms all we can state is that empirically those algorithms that maximise mutual information by also maximising the entropy explicitly exhibited more robustness. The proposed method does so in the most direct way which seems to favour even more its robustness but the exact reason remains to be proven theoretically in order to make a more formal statement.

---

> ### Author Response · Authors · 2022-11-17
> **Answer to Reviewer D75g [3/3]**
>
> * *How is the parameter of kernel band width selected? Or how the density is estimated?*
>
> This might be a common question for potential readers so thank you for creating the opportunity to let us explain it.  We included a longer discussion in the convergence analysis (Appendix F).
>
> From Theorem 2, we see that (5) is a consistent estimator of a lower bound of the mutual information as long as $h \in \mathcal{O}(k^{-1/(d+\varepsilon)})$, which usually happens when $h \to 1$. So a sensible value would be a value slightly smaller than 1, e.g. 0.99.
>
> However, for any constant $h$, equation (5) tends to a term that lower bounds the mutual information plus a constant that does not depend on the optimisation, so other values of $h$ can be employed when one cares about the optimisation and not the estimation. In practice, we observed that values between 0 and 1 work well. In fact, for the experiments with the von Mises—Fisher distribution we used $h = 0.1$ to maintain the same multiplicative value as CMC and SimCLR.

---

> > ### Comment · Reviewer_D75g · 2022-11-19
> > **Thank you for the response**
> >
> > The response and paper revision indeed help improve the overall quality. I appreciate the clarification on how Eq. (1) relates to the criteria in Zimmermann et al. (2021) and Von Kugelgen et al. (2021). However, the effectiveness of maximizing Eq. (1) still remains unclear despite the fact that it may unify results in prior works. As mentioned in Weakness (6), the validation of the proposed theoretical analysis could be made more compelling by using some synthetic experiments (as what is done in Zimmermann et al. (2021) and Von Kugelgen et al. (2021)). For example, it would be interesting to see whether maximizing Eq. (1) leads to comparable or better content/style separation results compared Von Kugelgen et al. (2021) or other baselines. Does the method recovers true latent component with simply affine ambiguity, as shown in Zimmermann et al. (2021)?
> >
> > In my opinion, the experiment validation is not consistent with the theoretical claims. Showing that the proposed algorithm is robust to hyperparameters does not corroborate the theoretical results on model identification in any way.

---

### Author Response · Authors · 2022-11-17
**Common message to all reviewers and the AC**

We would like to thank all reviewers for their valuable feedback. Thanks to their effort and constructive comments we were able to strengthen the paper, make it clearer, and even improve some of the results. We have answered each reviewer individually and with more details but would like to provide a summary of common concerns here.

A common comment was how one of our contributions (the fact that multi-view SSL methods are a lower bound on the mutual information) was already known, hence diminishing the novelty of our analysis. We have answered by providing more details based on each specific comment but in general, we acknowledge that methods based on InfoNCE (e.g., CMC and MoCo) were indeed known to be bounded by mutual information (Poole et al 2019). The analysis in (Poole et al 2019), however, assumes that the negative samples are independent and identically distributed and limits its applicability to InfoNCE (e.g., they do not hold for SimCLR). Conversely, our proposed analysis lifts some of the assumptions previously made. Furthermore, we include analyses encompassing other methods such as SimCLR, DeepCluser, SwAV, BYOL, or DINO. To the best of our knowledge, this was not possible before.

A comment made by some reviewers is that the proposed algorithm does not perform as well as algorithms like DINO. We clarified that the contribution of this work was not to provide a new benchmark-beating algorithm. After showing that many SSL methods try to maximise the mutual information, we noted that none of them did so in the most direct way. Hence we showed that with a simpler and direct algorithm one could achieve results comparable to other contrastive learning frameworks. Additionally, our algorithm resulted to be the most robust to changes in individual hyper-parameters such as the batch size and the number of epochs.

One reviewer was specifically concerned that our reported accuracy for BYOL did not match the accuracy reported in the original work (Grill et al. 2020). The reviewer is correct, the results we obtained with our implementation were inferior to those previously published. On the one hand, we are happy to report that after further hyper-parameter tuning we were able to reach, and even surpass, the results reported by Grill et al. (2020). On the other hand, what is important for our analysis is not the absolute accuracy but the speed of degradation when a single hyper-parameter is changed. Hence, even with the new results, the conclusion remains unchanged. Further, we note that Grill et al. (2020) did show that when reducing the batch size one can still maintain good accuracy by using gradient accumulation. However, in this experiment we are not testing the ability of an algorithm to perform well with small batches when re-doing hyper-parameter tuning or adding new strategies, we want to measure what happens if only 1 hyper-parameter (e.g., the batch size) is changed and the rest of the optimization is unchanged.

Finally, we would like to recapitulate the main contributions of our work.

* We provide an analysis of a few previously published works in SSL tying them together through the lens of information theory. In particular, they aim to maximise a lower bound on the mutual information. While this was known for InfoNCE-based algorithms, our analysis is applicable to more algorithms.
* We show that maximising a certain lower bound on the mutual information has the properties of being able to recover the true latent variables and to separate content from style, unifying the theory from (Zimmermann et al. 2021) and (Von Kügelgen et al. 2021).
* We observed that none of the existing algorithms maximises this mutual information lower bound in a direct way, so we proposed such an algorithm and evaluated its performance.
* We empirically show that different families of algorithms (contrastive vs reconstruction-based) exhibit different levels of robustness with respect to hyper-parameter choices. Namely, those that attempt to maximise the entropy directly show better robustness with respect to changes of single hyper-parameters such as batch size and training epochs.
* We showed that the proposed algorithm performs on par with existing contrastive algorithms while being simpler than previous works.

To conclude, we appreciate all the effort of the reviewers and the constructive comments and we believe that the revised version is a stronger work thanks to them. You may find the changes in red in the revised version. Thank you!

---

### Decision · Program_Chairs · 2023-01-20

**Decision:**

Reject

**Justification For Why Not Higher Score:**

Reviewers agree that the paper is not strong enough in terms of either theoretical analysis or empirical results.

**Justification For Why Not Lower Score:**

N/A

**Metareview: Summary, Strengths And Weaknesses:**

This manuscript tries to unify several multi-view self-supervised learning algorithms into the same maximum mutual information framework, based on which a new variant, using KDE for entropy estimation, is proposed.

Strength:

The authors did a good job connecting different multi-view SSL algorithms.

Weaknesses:

The reviewers think the paper does present insight on multi-view SSL algorithms, but the novel contribution is not crystal clear. The reviewers also found flaws in the experimental setup, which led to initial poor performance of the baselines. The authors shall take into account the reviewer comments to improve the paper.